# Hyperbolic Dataset Distillation

**Wenyuan Li**
Hokkaido University
wenyuan@lmd.ist.hokudai.ac.jp

**Guang Li**[*]
Hokkaido University
guang@lmd.ist.hokudai.ac.jp

**Keisuke Maeda**
Hokkaido University
maeda@lmd.ist.hokudai.ac.jp

**Takahiro Ogawa**
Hokkaido University
ogawa@lmd.ist.hokudai.ac.jp

**Miki Haseyama**
Hokkaido University
mhaseyama@lmd.ist.hokudai.ac.jp

## Abstract

To address the computational and storage challenges posed by large-scale datasets in deep learning, dataset distillation has been proposed to synthesize a compact dataset that replaces the original while maintaining comparable model performance. Unlike optimization-based approaches that require costly bi-level optimization, distribution matching (DM) methods improve efficiency by aligning the distributions of synthetic and original data, thereby eliminating nested optimization. DM achieves high computational efficiency and has emerged as a promising solution. However, existing DM methods, constrained to Euclidean space, treat data as independent and identically distributed points, overlooking complex geometric and hierarchical relationships. To overcome this limitation, we propose a novel hyperbolic dataset distillation method, termed HDD. Hyperbolic space, characterized by negative curvature and exponential volume growth with distance, naturally models hierarchical and tree-like structures. HDD embeds features extracted by a shallow network into the Lorentz hyperbolic space, where the discrepancy between synthetic and original data is measured by the hyperbolic (geodesic) distance between their centroids. By optimizing this distance, the hierarchical structure is explicitly integrated into the distillation process, guiding synthetic samples to gravitate towards the root-centric regions of the original data distribution while preserving their underlying geometric characteristics. Furthermore, we find that pruning in hyperbolic space requires only 20% of the distilled core set to retain model performance, while significantly improving training stability. Notably, HDD is seamlessly compatible with most existing DM methods, and extensive experiments on different datasets validate its effectiveness. To the best of our knowledge, this is the first work to incorporate the hyperbolic space into the dataset distillation process. The code is available at https://github.com/Guang000/HDD.

## 1 Introduction

Recently, deep neural networks (DNNs) have demonstrated outstanding performance across a wide range of tasks. However, the continuous performance improvement has led to increasingly large datasets, which in turn have escalated storage costs and computational demands, emerging as key

---

[*]Correspondence to: Guang Li <guang@lmd.ist.hokudai.ac.jp>

39th Conference on Neural Information Processing Systems (NeurIPS 2025).

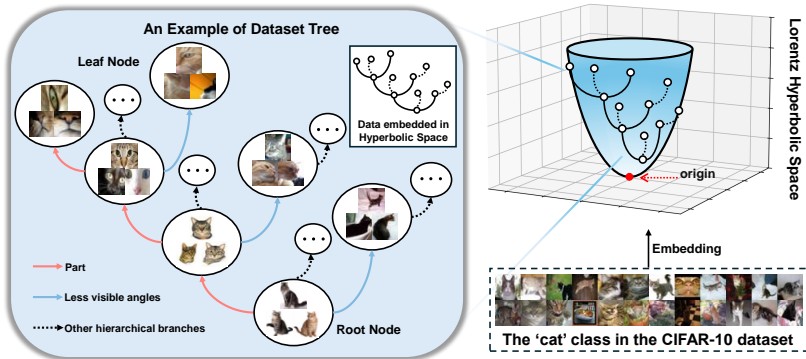

Figure 1: An example of hierarchical representation in hyperbolic space using the 'Cat' class from the CIFAR-10 Dataset. Hyperbolic space naturally encodes hierarchical structures. In this context, samples located near the root node often represent the category prototype more effectively, while those situated at higher hierarchical levels (closer to the leaf nodes) tend to contain noisier or specific information, such as object parts or less visible angles.

bottlenecks in the further advancement of deep learning. To address this issue, dataset distillation (DD) has been proposed [56]. By condensing the information of the original dataset, DD synthesizes a significantly smaller artificial dataset while striving to achieve comparable model performance. Beyond this, DD has also been widely applied in various domains, such as neural architecture search [68, 14, 42, 52], continual learning [19, 60], and privacy protection [13, 6, 30, 31].

To avoid the bi-level optimization problem of the DD methods, matching-based dataset distillation methods have been proposed. Currently, they can be broadly classified into three categories: gradient matching [69], trajectory matching [4, 14], and distribution matching [68, 64, 70]. The first two approaches can be collectively referred to as optimization-driven dataset distillation methods. Although these methods have achieved promising performance, their reliance on expensive optimization or nested gradients often incurs high computational costs, which hinders their scalability and broader application. In contrast, Zhao et al. proposed a distribution matching approach, which mitigates the need for expensive optimization by aligning the feature distributions encoded by neural networks from both the original and synthetic datasets, thereby reducing computational overhead [68]. Despite its advantages, distribution matching methods generally underperform optimization-driven approaches in terms of final model accuracy.

Distribution matching is typically divided into instance-level (point-wise) matching [55, 47] and moment matching [68, 64]. The central challenge lies in defining an effective metric to quantify the distributional discrepancy between the original and synthetic datasets. Point-wise matching is performed in Euclidean space by comparing feature representations using Mean Squared Error (MSE) on a per-sample basis. However, MSE primarily focuses on local alignment (e.g., pixel-wise similarity within samples) and tends to overlook the global semantic structure embedded in high-dimensional manifolds. In contrast, moment matching employs Maximum Mean Discrepancy (MMD) as a metric, which enables effective measurement of overall distribution differences in a Reproducing Kernel Hilbert Space (RKHS). Although both MSE and MMD attempt to reduce the distribution gap between original and synthetic datasets, they overlook a critical aspect: the hierarchical (or tree-like) structure inherent in dataset samples [59, 46], as illustrated in Figure 1. Under the hierarchy, the significance of samples varies—lower-level samples (closer to the root) tend to better represent the category prototype, whereas higher-level samples (closer to the leaves) often carry more irrelevant or noisy information [22, 20]. Treating all samples as independent and identically distributed (i.i.d.) when using MSE or MMD may thus degrade distillation performance.

To address the above-mentioned limitation, we introduce hyperbolic space as the distribution space for samples and propose a novel hyperbolic dataset distillation (HDD) method. Unlike Euclidean and Hilbert spaces, hyperbolic space is characterized by negative curvature, whose geometric constraints offer a continuous approximation of hierarchical tree-like structures, effectively capturing complex hierarchical relationships [12, 15]. In hyperbolic space, the centroid of a data distribution is the point that minimizes the total of squared hyperbolic distances to all sample points. Due to the unique

geometric properties of hyperbolic space, higher-level samples exert less influence on the centroid, naturally biasing it toward lower-level samples that are more representative of category prototypes. Nevertheless, the centroid still integrates the influence of all samples, which allows it to encode the overall geometric structure of the dataset. Based on this observation, we propose to match the distribution centroids of the original and synthetic datasets in hyperbolic space. This strategy aims to minimize distributional discrepancies, particularly concerning lower-level (prototype-like) samples, while also preserving the global geometric structure of the dataset [25]. The motivation of this study is that samples within a dataset contribute unequally to the overall representation depending on their hierarchical level, and the distillation process should be designed to reflect this imbalance. Notably, HDD is fully compatible with most existing dataset distillation methods. To the best of our knowledge, this is the first work to introduce hyperbolic space into the dataset distillation framework.

To summarize, our contributions are as follows:

- We propose hyperbolic dataset distillation (HDD), a novel method that incorporates hyperbolic geometry into dataset distillation to enable hierarchical sample weighting, effectively capturing semantic structures at multiple levels. Additionally, HDD aligns the global geometric distributions of the original and distilled datasets by matching their centroids in hyperbolic space.

- We analyze the contributions of samples at different hierarchical levels to the overall training loss, providing insights into their respective roles during distillation.

- Extensive experiments on diverse benchmarks, including Fashion-MNIST, SVHN, CIFAR-10, CIFAR-100, and TinyImageNet, demonstrate the effectiveness of our method. Additionally, our model also performs well in cross-architecture experiments.

- Furthermore, we apply hierarchical pruning to the original dataset by utilizing only the pruned subset for distribution alignment. Empirical results show that merely 20% of the original data suffices to preserve performance, underscoring the efficacy of hierarchical structuring within hyperbolic space.

## 2   Related Works

**Dataset Distillation.** Existing DD methods can be broadly categorized into three categories: gradient matching, trajectory matching, and distribution matching [34, 29, 63, 41]. Gradient matching [69, 67] seeks to preserve critical information by minimizing the discrepancy between the gradients induced by synthetic and original samples during model training. Trajectory matching [4, 14, 21, 9, 32, 33] achieves fine-grained knowledge transfer by aligning the training trajectories of network parameters. Distribution matching [68, 64, 70] improves the representational capacity of synthetic samples by aligning their statistical distributions with those of original data in feature or activation spaces. Recently, generative-based dataset distillation [18, 50, 51, 36, 37, 38, 39, 61] and decoupling optimization-based methods [62, 53, 49] have been proposed, accelerating advancements in the field of dataset distillation. In this work, we introduce hyperbolic space into dataset distillation by leveraging its inherent negative curvature to impose the tree-like hierarchy of the original dataset onto synthetic data, thereby offering a novel perspective to address the fundamental challenges in dataset distillation.

**Hyperbolic Machine Learning.** Hyperbolic space naturally encodes hierarchical data, which has attracted considerable interest in machine learning. It was first widely adopted in graph neural networks [5, 66, 17, 2] to more effectively capture hierarchical and complex graph structures. In computer vision and multimodal tasks, hyperbolic geometry has also been applied to metric learning [16, 45, 40], generation [8, 3], recognition [24], and segmentation [1]. As fully hyperbolic architectures have matured, hyperbolic-based vision methods have become increasingly sophisticated. In this work, we introduce hyperbolic space into dataset distillation for the first time, leveraging its hierarchical properties to assign differentiated weights to samples.

# 3 Method

## 3.1 Preliminaries

**Problem Definition.** Consider a large-scale original dataset $\mathcal{R} = \left\{(r_i^{\text{real}}, t_i^{\text{real}})\right\}_{i=1}^{|\mathcal{R}|}$, where $r_i^{\text{real}}$ represents the $i$-th sample instance from the original dataset, $t_i^{\text{real}}$ represents the corresponding label of the sample $r_i^{\text{real}}$ in the original dataset, and $|\mathcal{R}|$ is the total number of samples in the original dataset. The goal of dataset distillation is to construct a significantly smaller synthetic dataset $\mathcal{S} = \left\{(s_j^{\text{syn}}, t_j^{\text{syn}})\right\}_{j=1}^{|\mathcal{S}|}$, where $s_j^{\text{syn}}$ represents the $j$-th synthetic sample instance, $t_j^{\text{syn}}$ represents the corresponding label of the synthetic sample $s_j^{\text{syn}}$, and $|\mathcal{S}|$ is the total number of samples in the synthetic dataset, with $|\mathcal{S}| \ll |\mathcal{R}|$. Such that a model trained on $\mathcal{S}$ (denoted $\theta_{\text{syn}}$) exhibits performance comparable to one trained on $\mathcal{R}$ (denoted $\theta_{\text{real}}$) when evaluated on previously unseen samples. Formally, let $P_T$ denote the true data distribution and $\ell$ a loss function (e.g., cross-entropy), then the optimal synthetic dataset is obtained by minimizing the discrepancy in performance between $\theta_{\text{syn}}$ and $\theta_{\text{real}}$ as follows:

$$\mathcal{S}^\star = \arg \min_{E_{(p,t) \sim P_T}} \left\| \ell\left(\theta_{\text{syn}}(p), t\right) - \ell\left(\theta_{\text{real}}(p), t\right) \right\|, \tag{1}$$

where $(p, t)$ represents a sample pair drawn from the true data distribution $P_T$, with $p$ denoting the data instance and $t$ its corresponding label.

To tackle Eq. (1), previous optimization-based methods have primarily focused on two key strategies. One approach refines $\mathcal{S}$ through meta-learning, while the other aligns gradients or parameters between $\mathcal{S}$ and $\mathcal{R}$. Nevertheless, both strategies necessitate a bi-level optimization structure, which is computationally demanding due to the need for nested gradient computations. In contrast, DM [68] introduces distribution matching as a more efficient alternative by aligning the feature distributions between $\mathcal{S}$ and $\mathcal{R}$. Within this framework, the optimization of the condensed dataset is typically categorized into instance-level matching and moment matching. Instance level matching overlooks the global semantic structure of the data, making it a suboptimal choice. In contrast, moment matching is formulated as follows:

$$\mathcal{S}^\star = \arg \min_{\mathbb{E}_{\phi_Q \sim \mathcal{P}_{\phi_Q}}} \left\| \frac{1}{|\mathcal{R}|} \sum_{i=1}^{|\mathcal{R}|} \phi_Q(r_i^{\text{real}}) - \frac{1}{|\mathcal{S}|} \sum_{j=1}^{|\mathcal{S}|} \phi_Q(s_j^{syn}) \right\|^2, \tag{2}$$

where $\phi_Q \sim \mathcal{P}_{\phi_Q}$ represents a feature extractor randomly sampled from the distribution $\mathcal{P}_{\phi_Q}$ (typically instantiated by a randomly initialized DNN without the final linear classification layer).

**Hyperbolic Geometry.** In hyperbolic geometry, the $n$-dimensional hyperbolic space is formally defined as a Riemannian manifold $(M^n, g_K)$ endowed with a constant negative curvature $K < 0$, where $M^n$ denotes the underlying manifold and $g_K$ is the Riemannian metric that characterizes its geometric structure. To facilitate efficient and numerically stable computations, we adopt the Lorentz model $\mathbb{L}_K^n = (\mathcal{L}, g_L)$, which embeds the hyperbolic space into an $(n+1)$-dimensional Minkowski space. Here, $\mathcal{L}$ represents the set of points satisfying the constraint $\langle \mathbf{x}, \mathbf{x} \rangle_{\mathcal{L}} = 1/K$, and the metric tensor is given by $g_K = \text{diag}([-1, 1_n])$, the Lorentzian manifold can be defined as follows:

$$\mathcal{L} := \left\{ \mathbf{x} \in \mathbb{R}^{n+1} \ \middle| \ \langle \mathbf{x}, \mathbf{x} \rangle_{\mathcal{L}} = \frac{1}{K}, \ x_t > 0 \right\}. \tag{3}$$

Each point $\mathbf{x} \in \mathbb{L}_K^n$ can be expressed as a vector $\mathbf{x} = [x_t \ x_s]^T$, where $x_t > 0$ is referred to as the time component and $x_s \in \mathbb{R}^n$ as the spatial component. The Lorentzian inner product is defined as:

$$\langle \mathbf{x}, \mathbf{y} \rangle_{\mathcal{L}} := -x_t y_t + x_s^\top y_s. \tag{4}$$

Although several isometrically equivalent models exist in hyperbolic geometry, such as the Poincaré ball, the Klein model, and the upper half-space model, our work primarily utilizes the Lorentz model due to its analytical tractability and improved numerical behavior. The relevant details are explained in detail in Appendix A.

## 3.2 Hyperbolic Dataset Distillation for Distribution Matching

Given the original dataset $\mathcal{R} = \{(r_i^{\text{real}}, t_i^{\text{real}})\}_{i=1}^{|\mathcal{R}|}$ and the synthetic dataset for update $\mathcal{S} = \{(s_j^{\text{syn}}, t_j^{\text{syn}})\}_{j=1}^{|\mathcal{S}|}$ (where $|\mathcal{S}| \ll |\mathcal{R}|$), we first encode the data through a frozen pre-trained encoder $\phi$, generating corresponding feature vectors $v_i^{\text{real}}$ and $v_j^{\text{syn}}$ as follows:

$$v_i^{\text{real}} = \phi(r_i^{\text{real}}), v_j^{\text{syn}} = \phi(s_j^{\text{syn}}). \tag{5}$$

This process projects both original samples $r_i^{\text{real}}$ and synthetic samples $s_j^{\text{syn}}$ into Euclidean feature space parameterized by $\phi$. Subsequently, we map each sample from both the original and synthetic datasets to the hyperbolic space via the exponential map, yielding the hyperbolic embeddings $z_i^{\text{real}}$ and $z_j^{\text{syn}}$, respectively, as follows:

$$z_i^{\text{real}} = \exp_{p_0}(v_i^{\text{real}}) = \cosh(\sqrt{-K}\|v_i^{\text{real}}\|) \, p_0 \; + \; \sinh(\sqrt{-K}\|v_i^{\text{real}}\|) \, \frac{v_i^{\text{real}}}{\sqrt{-K}\|v_i^{\text{real}}\|}, \tag{6}$$

$$z_j^{\text{syn}} = \exp_{p_0}(v_j^{\text{syn}}) = \cosh(\sqrt{-K}\|v_j^{\text{syn}}\|) \, p_0 \; + \; \sinh(\sqrt{-K}\|v_j^{\text{syn}}\|) \, \frac{v_j^{\text{syn}}}{\sqrt{-K}\|v_j^{\text{syn}}\|}. \tag{7}$$

Here, $\|v\| = \sqrt{\langle v, v \rangle}$ denotes the norm induced by the Minkowski inner product, and $p_0$ represents the base point in the hyperbolic space, which is defined as:

$$p_0 = \left(\sqrt{-\frac{1}{K}}, 0, 0, \ldots, 0\right), \tag{8}$$

where $K < 0$ denotes the curvature of the hyperbolic space.

To facilitate subsequent analysis, we collect all hyperbolic embeddings of the original and synthetic datasets into two sets:

$$Z^{\text{real}} = \{z_i^{\text{real}}, t_i^{\text{real}}\}_{i=1}^{|\mathcal{R}|}, \quad Z^{\text{syn}} = \{z_j^{\text{syn}}, t_j^{\text{syn}}\}_{j=1}^{|\mathcal{S}|}. \tag{9}$$

Here, $Z^{\text{real}}$ and $Z^{\text{syn}}$ denote the sample points in the hyperbolic space corresponding to the real and synthetic samples, respectively. Unlike distribution matching methods in Euclidean space, in hyperbolic space, the distributional center of each embedded dataset is characterized by its Riemannian (Karcher) mean. We define their Riemannian means in the Lorentz model as:

$$\bar{z}^{\text{real}} = \arg\min_{z \in \mathbb{H}_K^n} \sum_{i=1}^{|\mathcal{R}|} d_L^2(z, z_i^{\text{real}}), \quad \bar{z}^{\text{syn}} = \arg\min_{z \in \mathbb{H}_K^n} \sum_{j=1}^{|\mathcal{S}|} d_L^2(z, z_j^{\text{syn}}), \tag{10}$$

where $z$ denotes a point in the Lorentzian hyperbolic space $\mathbb{L}_K^n$ over which the Riemannian mean is optimized, and the Lorentzian hyperbolic distance $d_L$ on the upper-sheet hyperboloid model is

$$d_L(m, n) = \frac{1}{\sqrt{-K}} \text{acosh}(-K \langle m, n \rangle_{\mathcal{L}}), \quad m, n \in \mathbb{L}_K^n, \tag{11}$$

and $\langle \cdot, \cdot \rangle_{\mathcal{L}}$ denotes the Minkowski inner product, as shown in Eq. (4).

To mitigate the extra computational overhead introduced by iterative procedures, we employ the centroid approximation approach proposed by Law et al. [25], which can be expressed as follows:

$$\mathbf{c} = \sqrt{-K} \cdot \frac{\bar{\mathbf{z}}}{\sqrt{|\langle \bar{\mathbf{z}}, \bar{\mathbf{z}} \rangle_{\mathcal{L}}| + \epsilon}}, \quad \text{where} \quad \bar{\mathbf{z}} = \frac{1}{n} \sum_{i=1}^n \mathbf{z}_i. \tag{12}$$

Here, $\mathbf{z}_i \in \mathbb{R}^{d+1}$ denotes the input vectors in Lorentzian hyperbolic space, $\bar{\mathbf{z}}$ is their Euclidean average. The curvature constant $K < 0$ reflects the negative curvature of the hyperbolic space. A small $\epsilon > 0$ is added for numerical stability.

Finally, we define the distribution matching loss as the Lorentzian hyperbolic distance between the two means as follows:

$$\mathcal{L}_{\text{Lhd}} = \lambda \, d_L(\bar{z}^{\text{real}}, \bar{z}^{\text{syn}}) = \frac{\lambda}{\sqrt{-K}} \text{acosh}\left(-K \langle \bar{z}^{\text{real}}, \bar{z}^{\text{syn}} \rangle_{\mathcal{L}}\right), \tag{13}$$

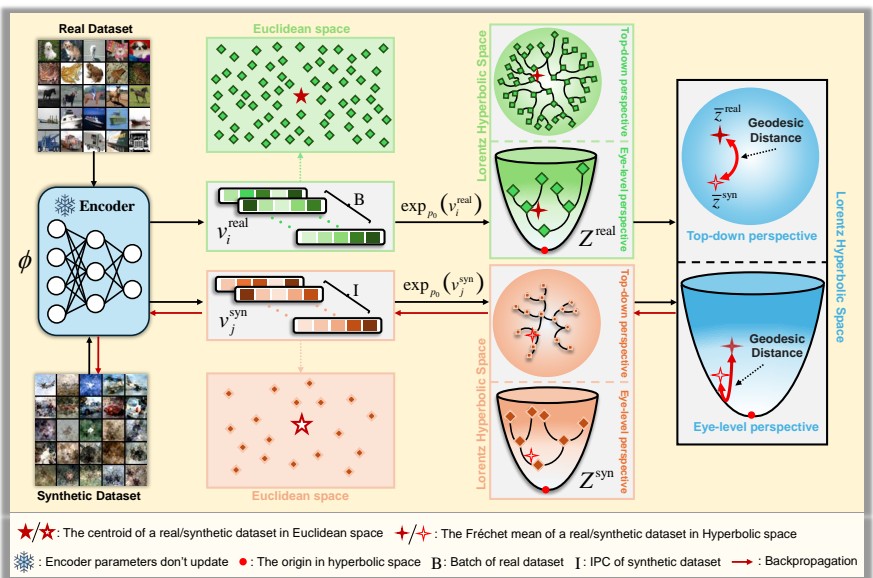

Figure 2: The framework of hyperbolic dataset distillation. The proposed method leverages exponential mapping to embed the dataset into hyperbolic space, enabling a hierarchical representation where samples at different levels are assigned varying weights to reflect their significance within the global geometry. Centroids of both the original and synthetic datasets are then computed in the hyperbolic space, and the geodesic distance between them is used to quantify the distributional discrepancy. This hyperbolic distance serves as a loss term to iteratively update the synthetic dataset, encouraging it to better align with the class-specific prototypes of the original data.

where $\lambda$ is the gradient scaling factor. In hyperbolic space, the centroid distribution is close to the origin, resulting in a very small distance between the centroids of the original dataset and the synthetic dataset. Additional parameters are required for amplification, as detailed in Appendix B.

Based on this loss, our objective in distribution matching is reformulated as minimizing the Lorentzian hyperbolic distance between the Riemannian means of the original and synthetic datasets:

$$\mathcal{S}^{\star} = \arg \min_{\mathbb{E}_{\phi_Q \sim \mathcal{P}_{\phi_Q}}} \left[ \lambda \, d_L\big(\bar{z}^{\text{real}}, \, \bar{z}^{\text{syn}}\big) \right]. \tag{14}$$

As illustrated in Fig. 2, the framework of HDD is presented. It is compatible with a broad range of existing distribution matching frameworks.

### 3.3 Loss Contribution of Samples at Different Levels

In hyperbolic space, samples embedded at lower levels tend to better represent category prototypes. When calculating the centroid, hyperbolic space can effectively assign different weights to relatively lower-level and higher-level samples, meaning their contributions to the centroid vary in influence. To gain explicit insight into how each sample influences the alignment between the original dataset $\mathcal{R}$ and the synthetic dataset $\mathcal{S}$ in hyperbolic space, we adopt a tangent space approximation centered at the origin $o \in \mathbb{L}_K^n$. Since the centroids of the sets ($\bar{z}^{\text{real}}$ and $\bar{z}^{\text{syn}}$) are near the origin, this approximation is reasonably effective. For $Z^{\text{real}}$ and $Z^{\text{syn}}$, respectively, define the hyperbolic radius (distance to the origin) of each sample as:

$$r_i = d_L(o, r_i^{\text{real}}), \qquad s_j = d_L(o, s_j^{\text{syn}}), \tag{15}$$

and let the corresponding normalized tangent vectors at the origin be

$$u_i = r_i^{\text{real}} - \cosh r_i \, o, \qquad v_j = s_j^{\text{syn}} - \cosh s_j \, o. \tag{16}$$

These vectors satisfy $u_i, v_j \in T_o\mathbb{L}_K^n$ (tangent space at the origin), i.e., they lie in the tangent space at the origin and satisfy $\langle u_i, o \rangle_L = \langle v_j, o \rangle_L = 0$.

To capture the radial influence of each sample, we introduce the scalar weight function (the derivation process is in Appendix C):

$$w(\mathbf{r}) = \frac{\sqrt{|K|}\, d}{\sinh(\sqrt{|K|}\, d)}, \tag{17}$$

which is strictly decreasing in $d$. $d$ represents the distance from the corresponding point to the reference point, which is defined as the origin in this context. This reflects that samples closer to the origin (i.e., with smaller hyperbolic norm) contribute more strongly to the Fréchet mean in the tangent space.

Under the tangent-space approximation of the Fréchet mean condition (i.e., the first-order optimality condition for the squared distance sum), the logarithmic maps of the centroids can be approximated as:

$$Log_o(\bar{z}^{\mathrm{real}}) \approx \sum_{i=1}^{|\mathcal{R}|} w(r_i)\, u_i, \qquad Log_o(\bar{z}^{\mathrm{syn}}) \approx \sum_{j=1}^{|\mathcal{S}|} w(s_j)\, v_j. \tag{18}$$

This yields the approximate loss function as the Euclidean distance between the two log-mapped centroids in the tangent space:

$$\mathcal{L}_{\mathrm{approx}} = d_L(\bar{z}^{\mathrm{real}}, \bar{z}^{\mathrm{syn}}) \approx \left\| \sum_{i=1}^{|\mathcal{R}|} w(r_i)\, u_i \; - \; \sum_{j=1}^{|\mathcal{S}|} w(s_j)\, v_j \right\|_{T_o \mathbb{L}_K^n}. \tag{19}$$

This formulation explicitly reveals the per-sample contribution to the overall loss: each sample affects the direction of the weighted log-mean, with its impact modulated by the scalar weight $w(r)$. Specifically, central samples (closer to the origin) receive higher weights, while peripheral samples (with larger hyperbolic radius) contribute less. This reflects a natural attenuation of influence in hyperbolic geometry and enhances stability by reducing the effect of outliers. Furthermore, we also explain this phenomenon from the perspective of gradients. For details, please refer to Appendix D.

## 4 Experiments

### 4.1 Experimental Setup

**Dataset.** We evaluated HDD on several standard benchmark datasets, including Fashion-MNIST [58], SVHN [43], CIFAR-10 [23], CIFAR-100 [23], and the larger-scale TinyImageNet [26]. Additionally, for hybrid architecture experiments, we utilized the ImageWoof subset of ImageNet [10], which features higher resolution images. Please refer to Appendix E for detailed information about the datasets used.

**Network Architectures.** For our primary experiments, we adopt the same convolutional network (ConvNet [27]) architecture as used in DC [69], DM [68], and IDM [70] to extract feature representations. This ConvNet consists of three sequential modules, each comprising a convolutional layer, instance normalization, a ReLU activation, and a stride-2 average pooling layer. To evaluate cross-architecture generalization, we follow the protocol in DM and conduct experiments using ConvNet, AlexNet, VGG11, and ResNet18 (The results can be found in Appendix F). For hybrid architecture experiments, we adopt the architectural configuration proposed in Dance [64].

**Implementation Details.** Our hyperparameter settings follow the design of the DM [68], IDM [70], and Dance [64] architectures. We adopt the differentiable siamese augmentation [67] enhancement method used in prior works. The synthetic dataset is learned using SGD. For DM with HDD, we train for 20,000 iterations, while for IDM with HDD and Dance with HDD, we train for 10,000 iterations. For all experiments, we set the batch size to 256. Additionally, for different experiments, we use distinct hyperbolic curvature $K$, gradient scaling factor $\lambda$, and synthetic image learning rate $r$, as detailed in Appendix G. All experiments are conducted on one RTX A6000 Ada GPU, except for Section 4.5.

### 4.2 Main Results

In the main results section, we established a comprehensive set of baseline methods to evaluate model performance. For core set selection approaches, we employed Random Selection [7], Herding [57], K-Center [48], and Forgetting [54]. Within the category of optimization-based methods, we incorporated

Table 1: Comparison of different methods on the FashionMNIST, SVHN, CIFAR10, and CIFAR100 datasets with IPC = 1, 10, and 50.

| Method | FashionMNIST | | | SVHN | | | CIFAR10 | | | CIFAR100 | | |
|---|---|---|---|---|---|---|---|---|---|---|---|---|
| IPC | 1 | 10 | 50 | 1 | 10 | 50 | 1 | 10 | 50 | 1 | 10 | 50 |
| Ratio (%) | 0.017 | 0.17 | 0.83 | 0.014 | 0.14 | 0.7 | 0.02 | 0.2 | 1 | 0.2 | 2 | 10 |
| Random | 51.4±3.8 | 73.8±0.7 | 82.5±0.7 | 14.6±1.6 | 35.1±4.1 | 70.9±0.9 | 14.4±2.0 | 26.0±1.2 | 43.4±1.0 | 4.2±0.5 | 14.6±0.5 | 30.0±0.4 |
| Herding | 67.0±1.9 | 71.1±0.7 | 71.9±0.8 | 20.9±1.3 | 50.5±3.3 | 72.6±0.8 | 21.5±1.2 | 31.6±0.7 | 40.4±0.6 | 8.4±0.3 | 17.3±0.3 | 33.7±0.5 |
| K-Center | 66.9±1.8 | 54.7±1.5 | 68.3±0.8 | 21.0±1.5 | 14.0±1.3 | 20.1±1.4 | 21.5±1.3 | 14.7±0.7 | 27.0±1.4 | 8.3±0.3 | 7.1±0.2 | 30.5±0.3 |
| Forgetting | - | - | - | 12.1±5.6 | 16.8±1.2 | 27.2±1.5 | 13.5±1.5 | 23.3±1.0 | 23.3±1.1 | 4.5±0.3 | 15.1±0.2 | 30.5±0.4 |
| DC [69] | 70.5±0.6 | 82.3±0.4 | 83.6±0.4 | 31.2±1.4 | 76.1±0.6 | 82.3±0.3 | 28.3±0.5 | 44.9±0.5 | 53.9±0.5 | 12.8±0.3 | 25.2±0.3 | - |
| DSA [67] | 70.6±0.6 | **84.6±0.3** | 88.7±0.3 | 27.5±1.4 | 79.2±0.5 | 84.4±0.4 | 28.8±0.7 | 52.1±0.5 | 60.6±0.5 | 13.9±0.3 | 32.3±0.3 | 42.8±0.4 |
| CAFE [55] | 77.1±0.9 | 83.0±0.4 | 84.8±0.4 | 42.6±3.3 | 75.9±0.6 | 81.3±0.3 | 30.3±1.1 | 46.3±0.6 | 55.5±0.6 | 12.9±0.3 | 27.8±0.3 | 37.9±0.3 |
| CAFE+DSA [55] | 73.7±0.7 | 83.0±0.3 | 88.2±0.3 | 42.9±3.0 | 77.9±0.6 | 82.3±0.4 | 31.6±0.8 | 50.9±0.5 | 62.3±0.4 | 14.0±0.3 | 31.5±0.2 | 42.9±0.2 |
| DCC [28] | - | - | - | 34.3±1.6 | 76.2±0.8 | 83.3±0.2 | 34.0±0.7 | 54.4±0.5 | 64.2±0.4 | 14.6±0.3 | 33.5±0.3 | 39.4±0.4 |
| G-VBSM [49] | - | - | - | - | - | - | - | 46.5±0.7 | 54.3±0.3 | 16.4±0.7 | 38.7±0.2 | 45.7±0.4 |
| DataDAM[47] | - | - | - | - | - | - | 32.0±1.2 | 54.2±0.8 | 67.0±0.4 | 14.5±0.5 | 34.8±0.5 | **49.4±0.3** |
| DM [68] | 70.7±0.6 | 83.4±0.1 | 88.1±0.6 | 21.9±0.4 | 72.8±0.3 | 82.6±0.3 | 26.4±0.3 | 48.5±0.6 | 62.2±0.5 | 11.4±0.3 | 29.7±0.3 | 43.0±0.4 |
| **DM with HDD** | 72.1±0.2 | 84.0±0.1 | **88.8±0.4** | 25.0±0.2 | 75.1±0.2 | 83.0±0.2 | 28.7±0.2 | 50.3±0.3 | 63.2±0.4 | 13.3±0.2 | 30.1±0.1 | 43.8±0.2 |
| IDM [70] | 77.4±0.3 | 82.4±0.2 | 84.5±0.1 | 65.3±0.3 | 81.0±0.1 | 85.2±0.3 | 45.2±0.5 | 57.3±0.3 | 67.2±0.1 | 22.1±0.2 | 44.7±0.3 | 46.5±0.4 |
| **IDM with HDD** | **78.5±0.2** | 83.8±0.2 | 86.4±0.3 | **67.8±0.2** | **84.0±0.2** | **87.6±0.1** | **47.0±0.1** | **61.3±0.1** | **69.7±0.2** | **25.3±0.2** | **45.4±0.1** | 48.9±0.3 |
| Whole Dataset | 93.5±0.1 | | | 95.4±0.1 | | | 84.8±0.1 | | | 56.2±0.3 | | |

DC [69], DSA [67], and DCC [28]. For distribution-matching methods, our baselines included CAFE [55], CAFE+DSA [55], DataDAM[47], as well as DM [68] and IDM [70]. Additionally, we have also considered the decoupling optimization method G-VBSM [49]. Detailed descriptions of these baseline methods are provided in Appendix H. For DM, IDM, and HDD, each experiment is conducted three times, and the mean and standard deviation are reported.

Table 1 presents a comparative evaluation of our method against prior approaches on Fashion-MNIST [58], SVHN [43], CIFAR-10 [23], and CIFAR-100 [23]. The results for TinyImageNet [26] are provided in Appendix I. IDM augmented with HDD, which exploits the hierarchical inductive bias of hyperbolic space, consistently outperforms the baseline IDM across all benchmarks. Under the IPC = 1 setting, IDM with HDD achieves classification accuracies of 78.5% on FashionMNIST (+1.1%), 67.8% on SVHN (+2.5%), 47.0% on CIFAR-10 (+1.8%), and 25.3% on CIFAR-100 (+3.2%), demonstrating its superiority in low-data regimes. With IPC = 10, the proposed method attains 61.3% accuracy on CIFAR-10, a 4.0% improvement over IDM. Under IPC = 50, it yields gains of 2.4%, 2.5%, and 2.4% on SVHN, CIFAR-10, and CIFAR-100, respectively. Furthermore, DM with HDD also exhibits notable enhancements relative to DM: on SVHN, accuracy increases by 3.1% (IPC = 1) and 2.3% (IPC = 10), and on CIFAR-10 (IPC = 1) by 2.3%. We present some of the distilled images in Appendix K.

### 4.3 Hierarchical Pruning

To validate the efficacy of hyperbolic-space-aware hierarchical pruning, we conducted the pruning experiments on CIFAR-10 (IPC = 10) by comparing DM with HDD against IDM with HDD across varying pruning rates. Specifically, given a batch of the original CIFAR-10 dataset $\mathcal{D} = \{(r_i, t_i, x_t^i)\}_{i=1}^{N}$, where $x_t^i$ denotes the time component of sample $i$, we sort all samples in descending order of $x_t^i$ and remove the top

Table 2: The distillation accuracy of CIFAR10 (IPC = 10) for different pruning ratios.

| Pruning Ratio | DM | DM with HDD | IDM with HDD |
|---|---|---|---|
| **95%** | 48.2±0.6 | 49.6±0.5 | 59.1±0.4 |
| **80%** | 48.7±0.2 | 50.2±0.2 | 60.3±0.3 |
| **50%** | 48.8±0.2 | 50.3±0.1 | 60.9±0.2 |
| **0%** | 48.5±0.6 | 50.3±0.3 | 61.3±0.1 |

$\alpha\%$ of samples exhibiting the highest time component, with pruning ratios $\alpha \in \{95\%, 80\%, 50\%\}$. Formally, the retained subset is defined as

$$\mathcal{D}' = \left\{ (r_i, t_i, x_t^i) \in \mathcal{D} \mid \operatorname{rank}(x_t^i) > \lceil \alpha N \rceil \right\}, \tag{20}$$

where $\operatorname{rank}(x_t^i)$ denotes the position of $x_t^i$ in the descending-sorted list.

Table 2 presents the matching accuracy after hierarchical pruning: both DM and DM with HDD require only 20% of the original training set to maintain performance, while IDM with HDD likewise preserves the vast majority of its performance with just 20% of data. This observation demonstrates that lower-level samples possess greater representativeness in hyperbolic space. However, we also observed that excessively small sample sizes still lead to performance degradation, indicating that higher-level samples also influence the centroid. Furthermore, Figs. 3-(a) and (b) depict the accuracy trajectories throughout the distillation process under various pruning ratios for HDD-DM

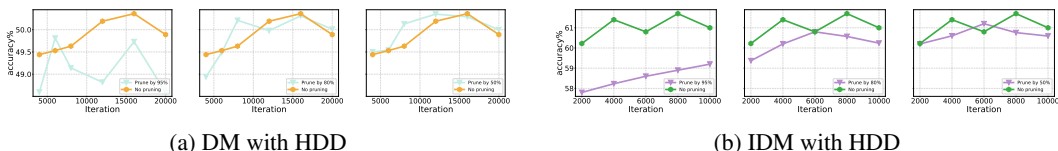

|          | (a) DM with HDD |          |          | (b) IDM with HDD |          |

Figure 3: Distillation accuracy variations of CIFAR-10 (IPC = 10) during the distillation process with different pruning rates.

## 4.4 Hybrid Architecture Experiment

Table 3: Comparison of different methods on the CIFAR10, CIFAR100, and ImageWoof datasets.

| Method | CIFAR10 | | | CIFAR100 | | | ImageWoof | |
|---|---|---|---|---|---|---|---|---|
| IPC | 1 | 10 | 50 | 1 | 10 | 50 | 1 | 10 |
| Ratio (%) | 0.02 | 0.2 | 1 | 0.2 | 2 | 10 | 0.11 | 1.10 |
| DATM [21] | 46.9±0.5 | 66.8±0.2 | 76.1±0.3 | 27.9±0.2 | 47.2±0.4 | 55.0±0.2 | - | - |
| RDED [53] | 23.5±0.3 | 50.2±0.3 | 68.4±0.1 | 19.6±0.3 | 48.1±0.3 | 57.0±0.1 | 18.5±0.9 | 40.6±2.0 |
| D⁴M [50] | - | 56.2±0.4 | 72.8±0.5 | - | 45.0±0.1 | 48.8±0.3 | - | - |
| IID (IDM) [11] | 47.1±0.1 | 59.9±0.2 | 69.0±0.3 | 24.6±0.1 | 45.7±0.4 | 51.3±0.4 | - | - |
| DSDM [35] | 45.0±0.4 | 66.5±0.3 | 75.8±0.3 | 19.5±0.2 | 46.2±0.3 | **54.0±0.2** | - | - |
| M3D [65] | 45.3±0.3 | 63.5±0.2 | 69.9±0.5 | 26.2±0.3 | 42.4±0.2 | 50.9±0.7 | - | - |
| Dance [64] | **47.2±0.3** | 70.2±0.2 | 76.3±0.1 | 26.2±0.2 | 49.7±0.1 | 52.8±0.1 | 27.1±0.2 | 46.2±0.2 |
| **Dance with HDD** | 46.8±0.3 | **70.8±0.2** | **77.1±0.2** | **27.7±0.3** | **50.2±0.2** | 53.9±0.1 | **27.6±0.2** | **46.6±0.1** |
| Whole Dataset | | 84.8±0.1 | | | 56.2±0.3 | | | 67.0±1.3 | |

To evaluate HDD's scalability, we ran additional experiments with the Hybrid Dance [64] architecture that alternates between cross-entropy and distribution matching optimization. We compared our proposed Dance with HDD method with leading distribution matching methods (IID [11], DSDM [35], and M3D [65]) as well as state-of-the-art approaches from other domains (DATM [21], RDED [53], D⁴M [50]), and the comprehensive experimental results are summarized in Table 3. On CIFAR-10 with IPC = 50, Dance with HDD improves over the original Dance by 0.8%. On CIFAR-100 with IPC = 1, it outperforms both Dance and M3D by 1.5%. Remarkably, at IPC = 10 on CIFAR-100, Dance with HDD is within 6% of training on the whole dataset. When scaling up to higher resolutions, our method still leads: on ImageWoof, it gains 0.5% at IPC = 1 and 0.4% at IPC = 10 compared to Dance. In addition, we also conducted our experiments on another hybrid architecture, DSDM [35]; please refer to Appendix J.

## 4.5 Runtime and GPU Memory Usage

We evaluate the computational overhead of DM with HDD relative to the baseline DM on the CIFAR-10 dataset. All experiments in this section are conducted on an RTX 4090. DM with HDD uses the same settings as DM (e.g., batch size, input image resolution), matching those in the main experiments. For runtime, we run 1,000 iterations and report the time per 100 iterations by dividing the total by 10.

Table 4: Runtime and GPU Memory Usage on CIFAR-10

| IPC | DM Runtime | DM with HDD Runtime | DM Memory | DM with HDD Memory |
|---|---|---|---|---|
| 1 | 4.9s | 6.7s | 3,522MiB | 3,522MiB |
| 10 | 5.0s | 6.8s | 3,626MiB | 3,632MiB |
| 50 | 5.4s | 7.1s | 3,888MiB | 3,922MiB |

As shown in Table 4, across IPC = 1/10/50 on CIFAR-10, adding HDD increases runtime from 4.9–5.4s to 6.7–7.1s, while GPU memory overhead is negligible. The effect is stable across IPC levels, indicating a modest, largely constant-time cost without inflating memory.

## 4.6 Ablation Study

We conducted an ablation study on different curvature values $K$ within the DM framework on CIFAR-10. As shown in the Table 5, although the curvature $K$ slightly affects the final accuracy, the variation is modest, and HDD consistently outperforms the Euclidean baseline. For example,

when IPC = 10, DM with HDD at curvatures $|K|$=1/3 and $|K|$=5 still outperforms plain DM by 1.1% and 1.5% percentage points, respectively. Note that the original DM is unaffected by curvature (its curvature is fixed at 0).

## 4.7 Discussion

The original CIFAR-10 data and the distilled synthetic sets with HDD were both projected onto the Poincaré ball for visualization; their centroids almost perfectly align. The essence of HDD lies in replacing the densely tree-structured distribution of the original dataset with a sparse tree-structured representation. As shown in Figs. 4-(a) and (c), although the number of samples in the synthetic dataset is significantly smaller, it still approximately captures

Table 5: Accuracy of DM with HDD at different curvature values.

| IPC | Method | $|K|$ | | | | | |
|---|---|---|---|---|---|---|---|
| | | 0 | 1/3 | 0.5 | 1 | 2 | 5 |
| 1 | DM | 26.4±0.3 | - | - | - | - | - |
| | DM with HDD | - | 27.0±0.2 | 28.8±0.3 | 28.7±0.2 | 27.6±0.2 | 28.6±0.2 |
| 10 | DM | 48.5±0.6 | - | - | - | - | - |
| | DM with HDD | - | 49.6±0.3 | 49.9±0.1 | 50.3±0.3 | 50.1±0.1 | 50.0±0.2 |
| 50 | DM | 62.2±0.5 | - | - | - | - | - |
| | DM with HDD | - | 63.0±0.3 | 63.1±0.1 | 63.2±0.4 | 63.1±0.2 | 62.7±0.1 |

the distributional trajectory of the original dataset. The synthetic dataset tends to be denser in regions where the original data is dense and sparser in regions where the original data is sparse. However, we also observe a tendency of the synthetic samples to concentrate closer to the root node (i.e., toward the center), as illustrated in Fig. 4-(b). Despite the presence of pronounced edge accumulation in the original dataset (i.e., a large number of samples located near the boundary), the synthetic samples are noticeably "attracted" toward the direction of the root node. As shown in Fig. 4-(d), although the synthetic dataset contains fewer samples overall, it exhibits a higher concentration of points near the root node compared to the original dataset.

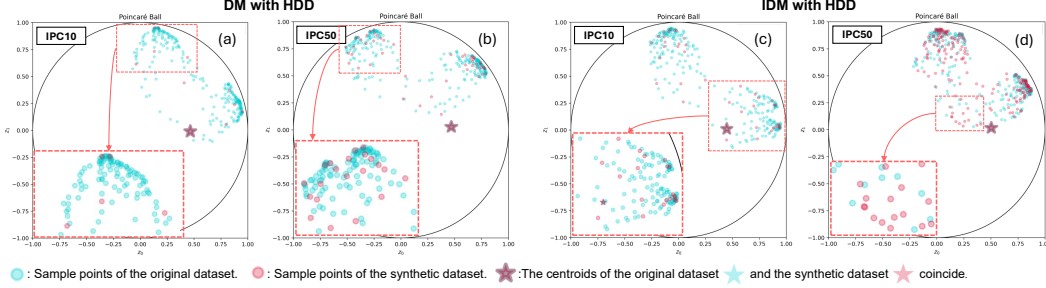

Figure 4: After distillation with DM with HDD and IDM with HDD, the distributions of the original and synthetic datasets in the Poincaré hyperbolic space are visualized.

## 5 Conclusion and Future Works

In this study, we introduce hyperbolic space into dataset distillation for the first time and propose a novel hyperbolic dataset distillation method, termed HDD. Leveraging the negative curvature of hyperbolic geometry, HDD effectively captures the hierarchical structure inherent in real-world datasets. By aligning the centroids of the original and synthetic datasets in hyperbolic space, we ensure that the synthetic data preserves the underlying geometric properties of the original data. Crucially, due to the varying influence of samples from different hierarchical levels on the centroid, the loss function naturally emphasizes contributions from lower-level (prototype) samples. This inductive bias enhances the preservation of class prototype distributions, thereby improving the quality of distillation. Currently, distribution metrics from information theory (e.g., KL divergence) and optimal transport theory (e.g., Wasserstein distance) have been extensively utilized in dataset distillation to enhance model performance. However, the application of these methods in hyperbolic dataset distillation remains unexplored, which presents a promising direction for future research to extend these methodologies into non-Euclidean-based dataset distillation.

## Acknowledgments

This research was supported in part by JSPS KAKENHI Grant Numbers JP23K11211, JP23K21676, JP24K02942, JP24K23849, and JP25K21218.

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

# Appendix

## A  Complementary Details of the Lorentz Hyperbolic Space

### A.1  Tangent Space $T_{\mathbf{x}}\mathcal{L}$

In the Lorentz model, hyperbolic space $\mathcal{L}$ is realized as a sheet of the two-sheeted hyperboloid in $\mathbb{R}^{n+1}$ with Minkowski metric. For any point $\mathbf{x} = [x_t; x_s] \in \mathcal{L}$, the tangent space captures all possible instantaneous directions at $\mathbf{x}$. It is defined by

$$T_{\mathbf{x}}\mathcal{L} = \left\{ \mathbf{v} \in \mathbb{R}^{n+1} \mid \langle \mathbf{x}, \mathbf{v} \rangle_{\mathcal{L}} = 0 \right\}. \tag{21}$$

This tangent space inherits the Lorentzian metric, and any tangent vector $\mathbf{v}$ has norm

$$\|\mathbf{v}\|_{\mathbf{x}} = \sqrt{\langle \mathbf{v}, \mathbf{v} \rangle_{\mathcal{L}}}, \tag{22}$$

which is strictly positive, ensuring that tangent vectors are purely spatial and providing the metric foundation for the exponential map.

### A.2  Exponential and Logarithm Maps

The exponential map pushes vectors in the tangent space onto the manifold, yielding a local Euclidean-like parametrization. Let $\kappa = \sqrt{-K}$. For $\mathbf{v} \in T_{\mathbf{x}}\mathcal{L}$, define

$$\exp_{\mathbf{x}}(\mathbf{v}) = \cosh\big(\kappa\|\mathbf{v}\|_{\mathbf{x}}\big)\,\mathbf{x} \; + \; \frac{\sinh\big(\kappa\|\mathbf{v}\|_{\mathbf{x}}\big)}{\kappa\|\mathbf{v}\|_{\mathbf{x}}}\,\mathbf{v}. \tag{23}$$

This formula satisfies $\exp_{\mathbf{x}}(0) = \mathbf{x}$ and ensures that the interpolation curve is a geodesic of constant curvature. The inverse (logarithm map) brings a point $\mathbf{y}$ back to the tangent space:

$$Log_{\mathbf{x}}(\mathbf{y}) = \frac{\mathrm{arccosh}\big(K\langle \mathbf{x}, \mathbf{y} \rangle_{\mathcal{L}}\big)}{\sqrt{-K\big(\langle \mathbf{x}, \mathbf{y} \rangle_{\mathcal{L}}\big)^2 - 1}}\,\big(\mathbf{y} - \langle \mathbf{x}, \mathbf{y} \rangle_{\mathcal{L}}\,\mathbf{x}\big). \tag{24}$$

### A.3  Bijection between the Lorentz and Poincaré Ball Models

For many applications (especially visualization), it is convenient to switch to the Poincaré ball. Given a Lorentz point $\mathbf{x} = [x_t; x_s]$, we map it to the unit ball $\|\mathbf{p}\| < 1$ via

$$\mathbf{p} = \frac{\kappa\, x_s}{1 + \kappa\, x_t}. \tag{25}$$

Conversely, for any $\mathbf{p} \in \mathbb{R}^n$ with $\|\mathbf{p}\| < 1$, set $\alpha = 1 - \|\mathbf{p}\|^2$ and recover

$$x_t = \frac{1 + \|\mathbf{p}\|^2}{\alpha}\,\frac{1}{\kappa}, \qquad x_s = \frac{2}{\alpha}\,\frac{\mathbf{p}}{\kappa}. \tag{26}$$

One verifies that the reconstructed $\mathbf{x}$ satisfies $-x_t^2 + \|x_s\|^2 = 1/K$ and $x_t > 0$.

## B  Centroid Convergence Toward the Origin

Given a finite sample $\{\mathbf{p}_i\}_{i=1}^N \subset \mathbb{L}_K^n$, define the Fréchet functional

$$F(\mathbf{p}) \;=\; \sum_{i=1}^N d_L^2\big(\mathbf{p}, \mathbf{p}_i\big), \tag{27}$$

Using the Riemannian logarithm $Log_{\mathbf{p}} \colon \mathbb{L}_K^n \to T_{\mathbf{p}}\mathbb{L}_K^n$, one obtains

$$\nabla F(\mathbf{p}) \;=\; -2\sum_{i=1}^N Log_{\mathbf{p}}(\mathbf{p}_i), \quad Log_{\mathbf{p}}(\mathbf{p}_i) \in T_{\mathbf{p}}\mathbb{L}_K^n, \tag{28}$$

so that the unique Fréchet mean $\mathbf{p}^*$ satisfies

$$\sum_{i=1}^{m} Log_{\mathbf{p}^*}(\mathbf{p}_i) = 0. \tag{29}$$

Since $\mathbb{L}_K^n$ has constant curvature $K < 0$, each map $\mathbf{p} \mapsto \| Log_{\mathbf{p}}(\mathbf{p}_i)\|^2$ is strictly convex along geodesics, ensuring a single global minimizer. We choose the origin $P_0$ to be the unique fixed point of a maximal compact subgroup of $Isom(\mathbb{L}_K^n)$, whose stabilizer is isomorphic to $\mathrm{O}(n)$. A comparison-theorem argument then shows

$$\| Log_{p_0}(\mathbf{p}_i)\| = d_L(p_0, \mathbf{p}_i) \geq \| Log_{\mathbf{p}}(\mathbf{p}_i)\| \quad \text{whenever } d_L(p_0, \mathbf{p}_i) \geq d_L(\mathbf{p}, \mathbf{p}_i), \tag{30}$$

forcing the solution of $\sum_i Log_{\mathbf{p}}(\mathbf{p}_i) = 0$ to lie radially closer to $p_0$ than the Euclidean centroid. Moreover, as $|K|$ increases, the lower bound on the second-derivative of $t \mapsto \| Log_{\gamma(t)}(\mathbf{p}_i)\|^2$ along any geodesic $\gamma$ grows, making this radial bias toward $p_0$ even more pronounced. This results in the centroids of both the original dataset and the synthetic dataset being biased towards $p_0$, while the distance between them is relatively small.

## C   Hierarchical Weight

In the hyperboloid model of constant sectional curvature $K < 0$, one introduces the scale parameter $\kappa = \sqrt{|K|}$ and radius $R = 1/\kappa$, so that the ambient space is

$$\mathcal{L} := \left\{ x \in \mathbb{R}^{n+1} \mid \langle x, x \rangle_{\mathcal{L}} = -R^2, \ x_0 > 0 \right\}, \tag{31}$$

where $\langle \cdot, \cdot \rangle_L$ denotes the Minkowski inner product of signature $(- + \cdots +)$. The geodesic distance between two points $p, q \in \mathbb{H}_K^n$ is given by

$$\begin{aligned} d_K(p, q) &= R \ arccosh\left(-\tfrac{1}{R^2}\langle p, q \rangle_L\right) \\ &= \frac{1}{\kappa} \ arccosh\left(-K \langle p, q \rangle_L\right). \end{aligned} \tag{32}$$

In particular, choosing the basepoint $o = (R, 0, \ldots, 0)$ and writing $r_i = d_K(o, x_i)$, one has

$$r_i = \frac{1}{\kappa} \ arccosh\left(-K \langle o, x_i \rangle_L\right), \tag{33}$$

$$\cosh(\kappa\, r_i) = \frac{-\langle o, x_i \rangle_L}{R^2}. \tag{34}$$

The logarithmic map at $o$ takes the form

$$Log_o(x_i) = \frac{\kappa\, r_i}{\sinh(\kappa\, r_i)} \left(x_i - \cosh(\kappa\, r_i)\, o\right). \tag{35}$$

Defining

$$w(r_i) = \frac{\kappa\, r_i}{\sinh(\kappa\, r_i)}, \tag{36}$$

$$u_i = x_i - \cosh(\kappa\, r_i)\, o, \tag{37}$$

one obtains

$$Log_o(x_i) = w(r_i)\, u_i. \tag{38}$$

## D   Gradient Contributions in the Lorentz Model of Hyperbolic Space

Given $N$ sample points $\{\mathbf{p}_i\}_{i=1}^{N} \subset \mathbb{L}_K^n$, their Fréchet mean (centroid) $\mu$ is defined by

$$\mu = \arg\min_{x \in \mathbb{H}_K^n} \sum_{i=1}^{N} d(x, p_i)^2, \tag{39}$$

so that the objective (loss) is

$$L(x) = \sum_{i=1}^{N} \left[ arcosh(-\langle x, p_i \rangle_L) \right]^2. \tag{40}$$

To study how a single point $p$ "pulls" on $x$, set

$$
\begin{aligned}
t &= -\langle x, p \rangle_L \\
&= \cosh\big(d(x,p)\big) \ge 1.
\end{aligned}
\tag{41}
$$

A standard derivation shows

$$
\nabla_x \, d(x,p)^2 = -2 \, \frac{arcosh(t)}{\sqrt{t^2-1}} \, \big(p + \langle x, p \rangle_L \, x\big),
\tag{42}
$$

and hence the magnitude of this pull is proportional to

$$
f(t) = \frac{arcosh(t)}{\sqrt{t^2-1}}.
\tag{43}
$$

**Asymptotic Behavior.**

*Near the "origin" ($t \to 1^+$).* Since $arcosh(t) \sim \sqrt{2(t-1)}$ and $\sqrt{t^2-1} \sim \sqrt{2(t-1)}$, we have

$$
f(t) = \frac{arcosh(t)}{\sqrt{t^2-1}} \longrightarrow 1.
\tag{44}
$$

Thus, points very close to $x$ exert almost the maximal pull of magnitude 1.

*Near the boundary ($t \to \infty$).* Using $arcosh(t) \sim \ln(2t)$ and $\sqrt{t^2-1} \sim t$ gives

$$
f(t) \sim \frac{\ln(2t)}{t} \longrightarrow 0,
\tag{45}
$$

so points very far from $x$ contribute almost no pull.

**Monotonicity.**

Differentiating

$$
f'(t) = \frac{\sqrt{t^2-1} - t \, arcosh(t)}{(t^2-1)^{3/2}},
\tag{46}
$$

we note that for all $t > 1$,

$$
\sqrt{t^2-1} < t \quad \text{and} \quad arcosh(t) > 1 \quad \Longrightarrow \quad t \, arcosh(t) > \sqrt{t^2-1},
\tag{47}
$$

so the numerator is negative while the denominator is positive. Hence

$$
f'(t) < 0 \quad \forall \, t > 1,
\tag{48}
$$

i.e. $f(t)$ is strictly decreasing on $(1, \infty)$.

Since $f(t)$ decreases from 1 to 0 as $t$ runs from $1^+$ to $\infty$, points closest to the current centroid $x$ exert the largest gradient pull, whereas points near the hyperbolic boundary (very far away) exert the smallest pull.

## E  Benchmark Datasets

We validate our hyperbolic dataset distillation method using six benchmark datasets: Fashion-MNIST [58], SVHN [43], CIFAR-10 [23], CIFAR-100 [23], Tiny ImageNet [26], and Image-Woof [10].

**FashionMNIST** is a drop-in replacement for the classic MNIST dataset, comprising 70,000 grayscale images of size $28 \times 28$ pixels across 10 apparel categories (e.g., T-shirt/top, sneaker) with a 60,000/1,000 train/test split.

**SVHN** contains approximately 600,000 real-world $32 \times 32$ RGB digit crops (0–9) collected from Google Street View images. It is partitioned into training (73,257), testing (26,032), and an extra set of 531131 samples for data augmentation.

**CIFAR-10** consists of 60,000 $32 \times 32$ color images evenly distributed over 10 object classes (airplane, car, bird, cat, deer, dog, frog, horse, ship, truck). There are five training batches of 10000 images each and one test batch, with exactly 1000 images per class.

**CIFAR-100** (building on CIFAR-10) contains 60,000 32 × 32 color images in 100 fine classes (600 images each) grouped into 20 coarse superclasses. Each fine class has a 500/100 train/test split, enabling hierarchical and fine-grained classification studies.

**Tiny ImageNet** is a subset of the ILSVRC-2012 challenge, selecting 200 classes and resizing all images to 64 × 64 pixels. It provides 100,000 images (500 train, 50 val, 50 test per class), offering a mid-scale benchmark between CIFAR and full ImageNet.

**ImageWoof** is a challenging subset of 10 visually similar dog breeds drawn from ImageNet (e.g., Beagle, Samoyed, Golden Retriever). It contains 9,025 training and 3,929 validation images, with optional noisy-label variants, and is commonly used to benchmark fine-grained recognition models.

## F   Cross-architecture Generalization

Cross-architecture generalization capability serves as a critical metric for evaluating the effectiveness of dataset distillation, where significant performance degradation across different architectures is deemed unacceptable. To assess this capability, we evaluated our method by testing its performance on ConvNet, AlexNet, VGG11, and ResNet18. As demonstrated in Table 6, both DM with HDD and IDM with HDD exhibit robust adaptability across diverse architectures. Compared with baseline DM and IDM methods, the HDD-enhanced approach demonstrates superior generalization strength and more stable performance while maintaining architectural compatibility.

Table 6: The distillation accuracy of CIFAR-10 (IPC = 10) for cross-architecture generalization.

| Model | ConvNet | AlexNet | VGG11 | ResNet18 |
|---|---|---|---|---|
| DSA [67] | 52.1±0.5 | 35.9±1.3 | 43.2±0.5 | 35.9±1.3 |
| KIP [44] | 47.6±0.9 | 24.4±3.9 | 42.1±0.4 | 36.8±1.0 |
| DM [68] | 48.9±0.6 | 38.8±0.5 | 42.1±0.4 | 41.2±1.1 |
| IDM [70] | 53.0±0.3 | 44.6±0.8 | 47.8±1.1 | 44.6±0.4 |
| **DM with HDD** | 50.3±0.3 | 46.3±0.4 | 45.7±0.3 | 40.2±0.4 |
| **IDM with HDD** | **61.3±0.1** | **57.2±0.3** | **58.6±0.4** | **56.8±0.3** |

## G   Hyperparameter Details

For different experiments, we use distinct hyperbolic curvature $K$, gradient scaling factor $\lambda$, and synthetic image learning rate $r$, as shown in Table 7 and Table 8. For the hyperbolic curvature $K$, we set it between $0.2$ and $3$. For the gradient scaling factor $\lambda$, we refer to the loss in Hilbert space and ensure that the hyperbolic distance loss maintains the same order of magnitude as the Hilbert space loss through $\lambda$. We make minor adjustments to the synthetic image learning rate $r$ while respecting the original method.

## H   Details of Baseline Methods

**Dataset Condensation (DC)** [69] achieves this objective by learning a synthetic dataset that, when used alongside the large dataset to train a deep network, results in comparable weight gradients.

**Differentiable Siamese Augmentation (DSA)** [67] enables learning synthetic training sets by applying identical random transformations to both real and synthetic data during training while supporting gradient backpropagation through differentiable augmentations.

**Dataset Condensation with Contrastive signals (DCC)** [28] enhances dataset condensation by matching summed gradients across all classes (unlike class-wise matching in DC) and optimizing synthetic data with contrastive signals. It stabilizes training via kernel velocity tracking and bi-level warm-up, improving fine-grained classification.

**Condense dataset by Aligning FEatures (CAFE)** [55] condenses data by aligning layer-wise features between real and synthetic data, explicitly encoding discriminative power into synthetic clusters, and adaptively adjusting SGD steps via a bi-level optimization scheme.

Table 7: Hyperparameter details of DM with HDD and IDM with HDD.

| Dataset | IPC | DM with HDD | | | IDM with HDD | | |
|---|---|---|---|---|---|---|---|
| | | $-1/K$ | $\lambda$ | $r$ | $-1/K$ | $\lambda$ | $r$ |
| FashionMNIST | 1 | 1 | 20 | 1 | 2 | 40 | 0.5 |
| | 10 | 1 | 40 | 1 | 2 | 60 | 1 |
| | 50 | 1 | 60 | 1 | 2 | 80 | 0.2 |
| SVHN | 1 | 1 | 10 | 1 | 2 | 120 | 0.5 |
| | 10 | 1 | 50 | 1 | 2 | 120 | 1 |
| | 50 | 1 | 100 | 1 | 2 | 120 | 0.2 |
| CIFAR 10 | 1 | 1 | 1 | 1 | 3 | 80 | 0.5 |
| | 10 | 1 | 20 | 1 | 3 | 100 | 1 |
| | 50 | 1 | 80 | 1 | 3 | 120 | 0.2 |
| CIFAR 100 | 1 | 1 | 10 | 1 | 2 | 60 | 0.5 |
| | 10 | 2 | 100 | 1 | 2 | 80 | 0.2 |
| | 50 | 2 | 120 | 1 | 2 | 100 | 0.6 |
| TinyImageNet | 1 | - | - | - | 2 | 80 | 0.5 |
| | 10 | - | - | - | 2 | 100 | 0.5 |
| | 50 | - | - | - | 2 | 120 | 0.6 |

Table 8: Hyperparameter details of Dance with HDD.

| Dataset | IPC | DM with HDD | | |
|---|---|---|---|---|
| | | $-1/K$ | $\lambda$ | $r$ |
| CIFAR-10 | 1 | 1.8 | 20 | 0.02 |
| | 10 | 0.2 | 40 | 0.2 |
| | 50 | 2 | 60 | 0.5 |
| CIFAR-100 | 1 | 2 | 40 | 0.02 |
| | 10 | 1.5 | 80 | 0.1 |
| | 50 | 2 | 120 | 0.5 |
| ImageWoof | 1 | 0.6 | 100 | 0.1 |
| | 10 | 0.5 | 120 | 0.1 |

**Dataset Distillation with Attention Matching (DataDAM)**[47] generates synthetic images by aligning the spatial attention maps of real and synthetic data, produced across various layers of a set of randomly initialized neural networks.

**Distribution Matching (DM)** [68] is the first to use maximum mean discrepancy to optimize synthetic data to match the distribution of the original data.

**Improved Distribution Matching (IDM)** [70] enhances DM by addressing feature imbalance through Partitioning and Expansion augmentation, and correcting invalid MMD estimation using enriched semi-trained model embeddings and class-aware distribution regularization, resulting in more accurate feature alignment and improved performance.

**Generalized Various Backbone and Statistical Matching (G-VBSM)** [49] is a novel framework for generalized dataset condensation, comprising three key components: data densification enhances intra-class diversity by ensuring linear independence within each class; generalized statistical matching captures patch- and channel-level convolutional statistics without gradient updates for effective synthesis; and generalized backbone matching enforces consistency across diverse backbones, boosting generalization. Together, they enable efficient and robust generalized matching.

**Difficulty-Aligned Trajectory Matching (DATM)** [21] dynamically adjusts the difficulty of synthetic data (matching the early or late training trajectories of the teacher network) to adapt to the scale of the synthetic dataset—small datasets correspond to simple modes (early trajectories), while large

datasets correspond to complex modes (late trajectories). This approach achieves lossless dataset distillation for the first time.

**Realistic, Diverse, and Efficient Dataset Distillation (RDED)** [53] is a non-optimization-based dataset distillation method that enhances realism by cropping realistic patches from original images and improves diversity by stitching these patches into new synthetic images, achieving high efficiency and superior performance on large-scale, high-resolution datasets.

**Dataset Distillation via Disentangled Diffusion Model (D$^4$M)** [50] leverages a disentangled diffusion model with a novel training-time matching strategy to efficiently distill high-resolution, realistic datasets while improving cross-architecture generalization and reducing computational costs.

**Inter-sample and Inter-feature Relations in Dataset Distillation (IID)** [11] introduces two key constraints to improve distribution matching: a class centralization constraint to enhance intra-class feature clustering, and a covariance matching constraint to accurately align feature distributions by considering both mean and covariance, even with limited synthetic samples.

**Diversified Semantic Distribution Matching (DSDM)** [35] distills datasets by aligning the semantic distributions—represented as Gaussian prototypes and covariance matrices—of distilled data with those of original data.

**Minimizing the Maximum Mean Discrepancy (M3D)** [65] enhances DM-based dataset condensation by aligning not only the first but also higher-order moments of feature distributions through kernel-based Maximum Mean Discrepancy, enabling more accurate distribution matching with theoretical guarantees and strong performance across diverse datasets.

**Dual-view distribution AligNment for dataset CondEnsation (DANCE)** [64] introduces a dual-view approach to dataset condensation by leveraging expert models: it performs pseudo long-term distribution alignment via a convex combination of initialized and trained models to align inner-class distributions without persistent training, and applies distribution calibration using expert models to mitigate inter-class distribution shift and preserve class boundaries.

## I  Results on TinyImageNet

Table 9: Comparison on TinyImageNet with different IPCs.

| Method | IPC = 1 (0.2%) | IPC = 10 (2%) | IPC = 50 (10%) |
|---|---|---|---|
| Random [7] | 1.4±0.1 | 5.0±0.2 | 15.0±0.4 |
| Herding [57] | 2.8±0.2 | 6.3±0.2 | 16.7±0.3 |
| K-Center [48] | 1.6±0.2 | 5.1±0.1 | 15.0±0.3 |
| Forgetting [54] | 1.6±0.2 | 5.1±0.3 | 15.0±0.1 |
| DC [69] | 5.3±0.1 | 12.9±0.1 | 12.7±0.4 |
| DSA [67] | 5.7±0.1 | 16.3±0.2 | 5.1±0.2 |
| DataDAM [47] | 8.3±0.4 | 18.7±0.3 | **28.7±0.3** |
| MTT [4] | 6.2±0.4 | 17.3±0.2 | 26.5±0.3 |
| IDM [70] | 10.1±0.2 | 21.9±0.6 | 26.9±0.2 |
| **IDM with HDD** | **11.9±0.2** | **22.4±0.3** | 27.8±0.3 |
| Whole Dataset | | 37.6±0.6 | |

We compare IDM with HDD against DC [69] , DSA [67], DataDAM [47], MTT [4], and IDM [70] on TinyImageNet, as shown in Table 9. Our method achieves superior performance at both IPC = 1 and IPC = 10. Furthermore, compared to IDM, IDM with HDD demonstrates improvements of 1.8%, 0.5%, and 0.9% at IPC = 1, IPC = 10, and IPC = 50, respectively.

## J  Experiments on an Alternative Hybrid Architecture (DSDM)

We evaluated the performance of HDD on the hybrid architecture DSDM [35] using the CIFAR-10 dataset. As shown in Table 10, when IPC = 1, DSDM with HDD achieved a 2.6% performance gain

compared to the original DSDM; when IPC = 10, DSDM with HDD showed an improvement of 0.8%.

Table 10: Accuracy comparison of DSDM with/without HDD.

| Method | IPC | Accuracy (%) |
|---|---|---|
| DSDM | 1 | 43.8 ± 0.2 |
| **DSDM with HDD** | 1 | **46.4 ± 0.3** |
| DSDM | 10 | 65.8 ± 0.3 |
| **DSDM with HDD** | 10 | **66.6 ± 0.4** |
| DSDM | 50 | 75.8 ± 0.2 |
| **DSDM with HDD** | 50 | **76.0 ± 0.2** |

# K Visualization of Distilled Images

We showcased a portion of the synthetic dataset distilled through HDD. Figure 5 displays the FashionMNIST samples synthesized using the DM with HDD at IPC = 50, while Figure 6 shows the analogous SVHN outputs under identical conditions. Figures 3 and 4 correspond to CIFAR-10: Figure 7 (a) and (b) depict the IDM with HDD results at IPC = 1 and IPC = 10, respectively, and Figure 8 demonstrates the IPC = 50 case. Figure 9 extends this analysis to CIFAR-100, presenting IDM with HDD distillations at IPC = 1 (a), IPC = 10 (b), and IPC = 50 (c). Finally, Figure 10 illustrates the ImageWoof distilled samples obtained via the Dance with HDD at IPC = 1 (a) and IPC = 10 (b).

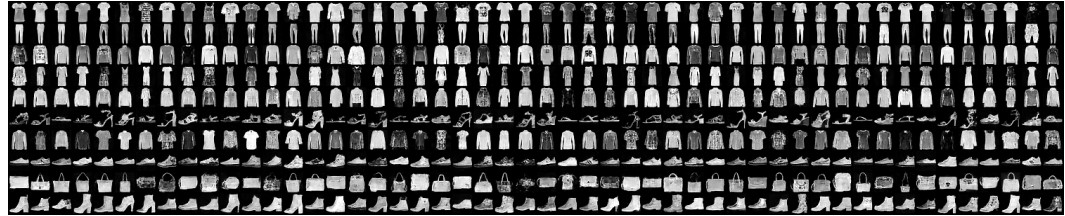

Figure 5: The distilled images of FashionMNIST with IPC = 50 using DM with HDD.

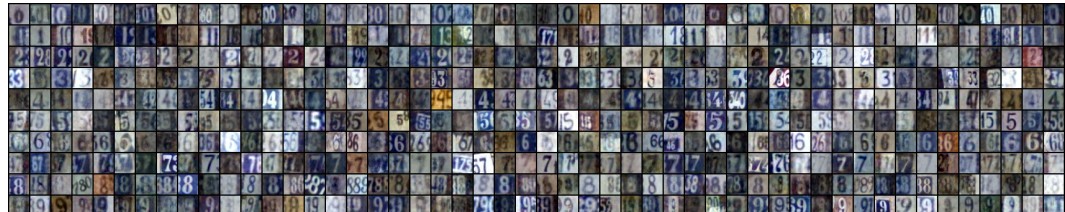

Figure 6: The distilled images of SVHN with IPC = 50 using DM with HDD.

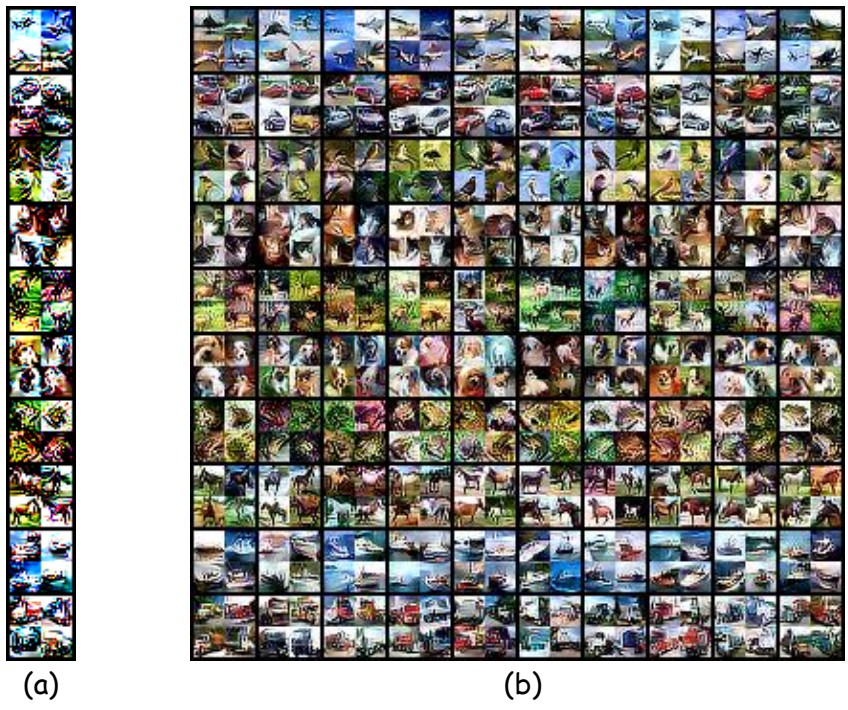

(a)          (b)

Figure 7: The distilled images of CIFAR-10 with IPC = 1 (a) and IPC = 10 (b) using IDM with HDD.

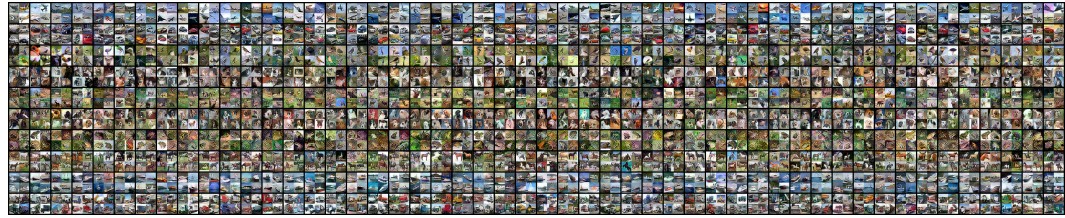

Figure 8: The distilled images of CIFAR-10 with IPC = 50 using IDM with HDD.

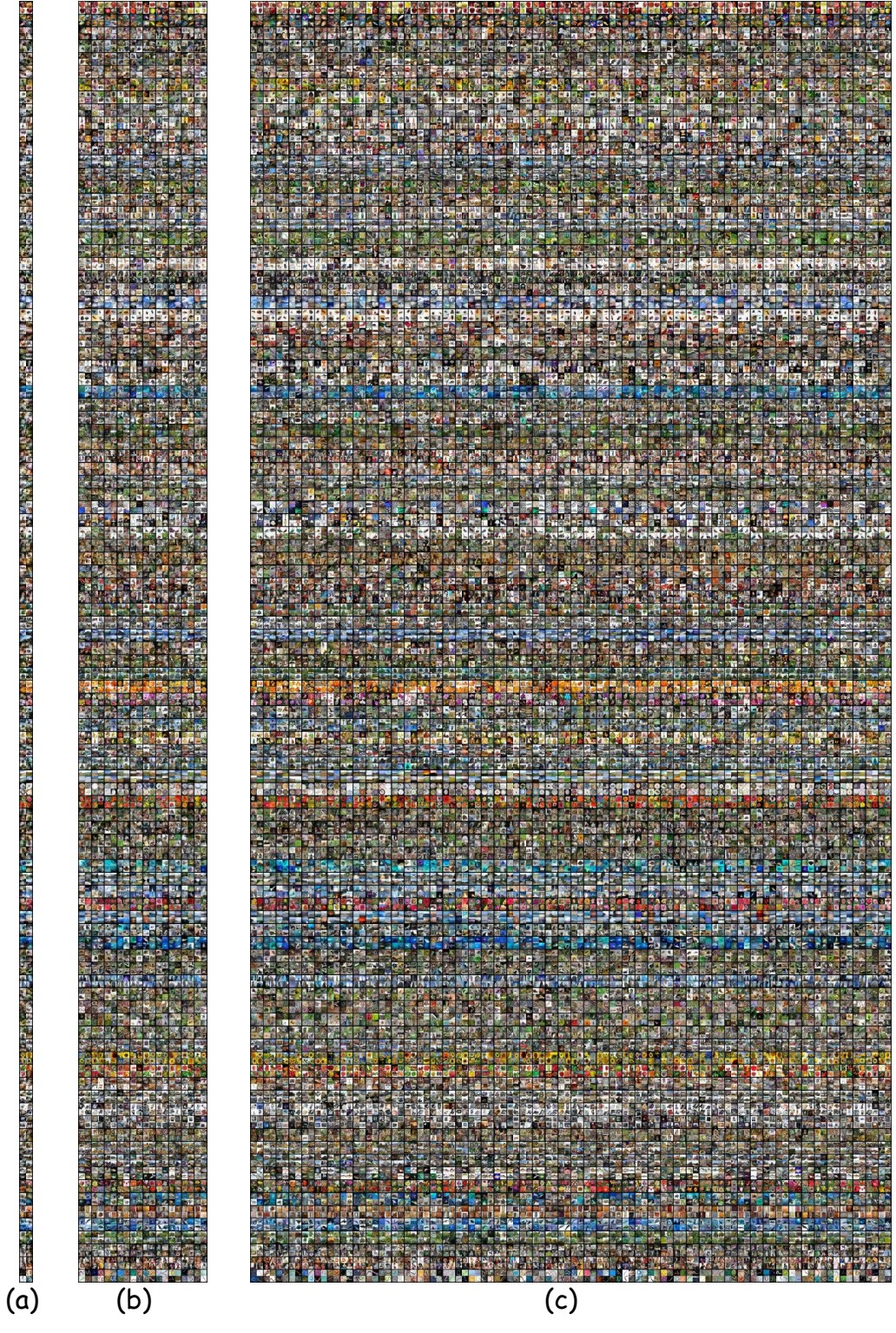

(a)          (b)                                    (c)

Figure 9: The distilled images of CIFAR-100 with IPC = 1 (a), IPC = 10 (b), and IPC = 50 (c) using IDM with HDD.

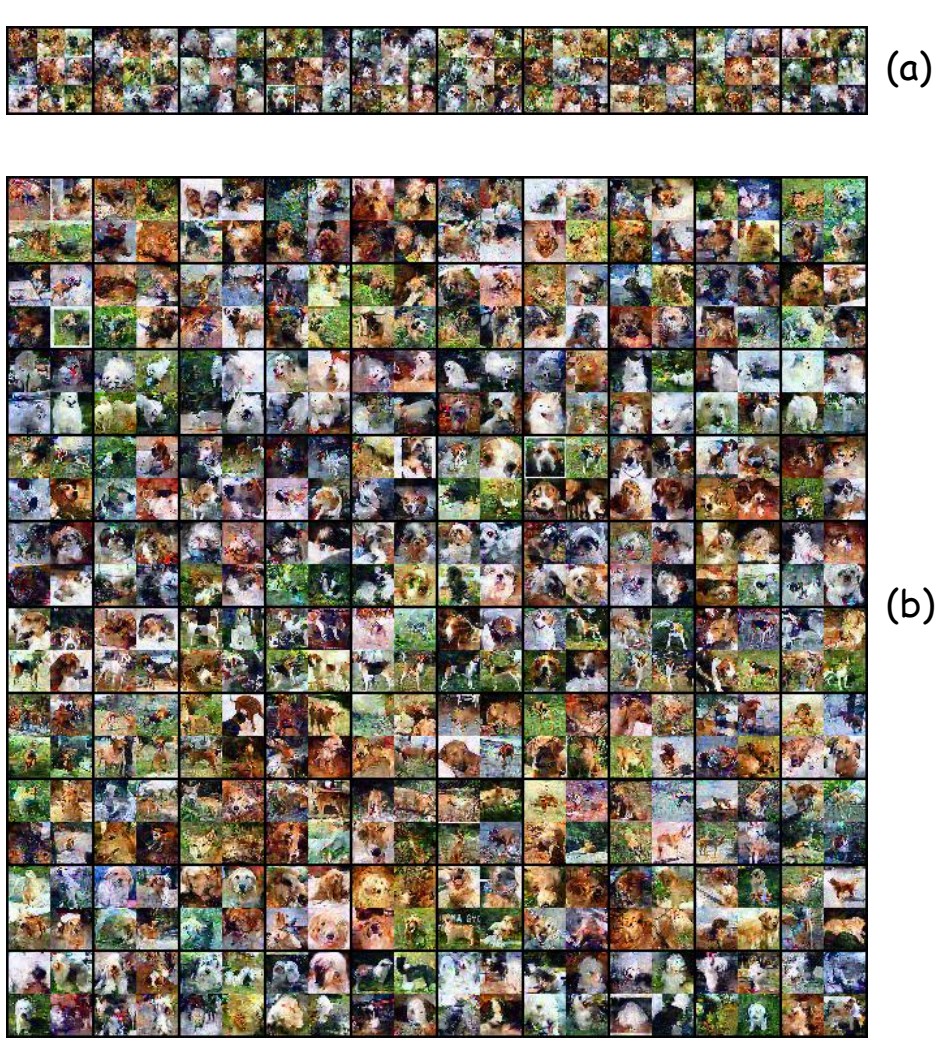

(a)

(b)

Figure 10: The distilled images of ImageWoof with IPC = 1 (a) and IPC = 10 (b) using Dance with HDD.

