# OpenReview forum: "Hyperbolic Dataset Distillation"
_NeurIPS.cc/2025/Conference — NeurIPS 2025 poster_

### Official Review · Reviewer_arwV · 2025-06-22

**Clarity:** 3
**Significance:** 3
**Originality:** 3
**Rating:** 4
**Confidence:** 4

**Summary:**

This paper investigates an issue with dataset distillation, where synthesized data in distribution matching method are trained under the setting of independent and identical distribution and fail to capture the complex geometric and hierarchical relationship. To address this problem, the authors propose a new dataset distillation strategy called Hyperbolic Dataset Distillation method (HDD). HDD embeds data into hyperbolic space and optimizes geodesic distance between the centroids of synthetic and original data, naturally capturing tree-like and hierarchical relationships. This approach allows hierarchical pruning and it is compatible with most existing distribution matching methods.

**Questions:**

1. How is the curvature associated with sample $i$ computed?
2. The authors mention that distribution matching methods improve efficiency, but optimization-based methods typically achieve higher performance. Is it feasible to apply HDD to optimization-based methods? If so, could the authors provide experimental results?
3. Could the authors provide more concrete evidence that the performance gains from HDD are indeed due to the presence of hierarchical structure in the original data? For the standard benchmark datasets used in the paper, the assumption of hierarchical relationships is not verified.

**Ethical Concerns:**

["NO or VERY MINOR ethics concerns only"]

**Final Justification:**

I thank the authors for their detailed and thoughtful rebuttal. The response fully resolves my main concern. I believe the paper is good enough to be accepted, and I raised my score accordingly.

**Limitations:**

In Sec. 5, the authors mention a limitation that more advanced distribution alignment objectives, such as KL divergence, are not explored in hyperbolic space, which could potentially improve performance further.

**Quality:**

3

**Strengths And Weaknesses:**

**Strengths**

This paper is well-motivated, and well-organized. The authors provide comprehensive experiments on various datasets such as CIFAR-10, CIFAR-100, TinyImageNet. They provide insightful analysis of the experimental results and a detailed method description.

**Weakness**
1. The encoder used to extract features before hyperbolic embedding. This may lead to the loss of important hierarchical information or cause feature collapse, where synthetic samples concentrate around the origin without preserving meaningful structure.
2. The effectiveness depend on tuning of curvature $K$ and scaling factor $\lambda$.
3. Some definitions are not clear: (a) In line 265, the definition of $K_i$ is ambiguous; (b) The correct presentation of Eq. (1) should be: $ \mathcal{S}^* = \arg \min_{\mathcal{S}} E_{(x, t) \sim P_T} || \ell(\theta_{\text{syn}}(x), t) - \ell(\theta_{\text{real}}(x), t) || $. Eq. (2) contains a similar mistake.
4. Some notations are abused. For example, the authors define $x$ as data instance, $t$ as label and $s$ as synthetic data with super script $syn$ in Sec 3.1 Problem Definition. However, in Sec 3.1 Hyperbolic Geometry, $x_t$ and $x_s$ are redefined as time component and spatial component. It is recommended to revise the notation to ensure consistency and clarity.

---

> ### Author Rebuttal · Authors · 2025-07-31
>
> ## Response to Reviewer arwV
>
> We sincerely appreciate your thoughtful review and constructive comments on our submission. Below, we provide detailed responses to your concerns.
>
> ---
>
> ### **Clarification 1 - Frozen Encoder: No Feature Collapse, Robust at Extreme Curvatures**
>
> We extract image features using a frozen network, whose weights remain unchanged during distillation; therefore, **the features do not collapse into trivial representations** (i.e., the synthesized samples are not forced to cluster around the origin).
> Moreover, our extensive experiments confirm that no feature collapse has been observed to date. Finally, we tested extreme values of $\|K\|$, and the results show that HDD maintains robust performance even in these edge cases, as presented below:
>
> | IPC   | Method       | No Curvature     | $\|K\|$ = 0.01 (Low Curvature) | $\|K\|$ = 100 (High Curvature) |
> |-------|--------------|------------------|----------------------------|----------------------------|
> | 1     | DM           | 26.4 ± 0.3       | -                          | -                          |
> | 1     | DM + HDD     | -                | 26.6 ± 0.2                 | 27.8 ± 0.2                 |
> | 10    | DM           | 48.5 ± 0.6       | -                          | -                          |
> | 10    | DM + HDD     | -                | 48.6 ± 0.3                 | 49.9 ± 0.2                 |
> | 50    | DM           | 62.2 ± 0.5       | -                          | -                          |
> | 50    | DM + HDD     | -                | 62.5 ± 0.3                 | 63.1 ± 0.1                 |
>
> We observe that when the curvature $\|K\|$ is very small, the performance gap between HDD and the original DM shrinks, which aligns with the theoretical expectation that hyperbolic space degenerates into Euclidean space as the curvature approaches zero. Conversely, when $\|K\|$ becomes very large, our method remains effective.
>
> ---
>
> ### **Clarification 2 – Ablation on Curvature $K$ and Loss-Weight $\lambda$**
>
> We conducted an ablation study on different curvature values $K$ within the DM framework on CIFAR-10; the results are presented below:
>
> ---
>
> | IPC   | Method       | $\|K\|$ = 1/3       | 0.5             | 1               | 2               | 5               |
> |-------|--------------|----------------|------------------|------------------|------------------|------------------|
> | 1     | DM           | 26.4 ± 0.3     | 26.4 ± 0.3      | 26.4 ± 0.3      | 26.4 ± 0.3      | 26.4 ± 0.3      |
> |       | DM + HDD     | 27.0 ± 0.2     | 28.8 ± 0.3      | 28.7 ± 0.2      | 27.6 ± 0.2      | 28.6 ± 0.2      |
> | 10    | DM           | 48.5 ± 0.6     | 48.5 ± 0.6      | 48.5 ± 0.6      | 48.5 ± 0.6      | 48.5 ± 0.6      |
> |       | DM + HDD     | 49.6 ± 0.3     | 49.9 ± 0.1      | 50.3 ± 0.3      | 50.1 ± 0.1      | 50.0 ± 0.2      |
> | 50    | DM           | 62.2 ± 0.5     | 62.2 ± 0.5      | 62.2 ± 0.5      | 62.2 ± 0.5      | 62.2 ± 0.5      |
> |       | DM + HDD     | 63.0 ± 0.3     | 63.1 ± 0.1      | 63.2 ± 0.4      | 63.1 ± 0.2      | 62.7 ± 0.1      |
>
> Although $\|K\|$ slightly affects the final accuracy, **the variation is modest, and HDD consistently surpasses the Euclidean baseline**.
>
> ---
>
> Unlike the curvature hyperparameter, the inter‑centroid distances in hyperbolic space are extremely small because of its unique geometry, so we introduce a scaling factor. **As long as this factor stays within a reasonable range, it exerts little influence on the final results.** We choose its value empirically: by inspecting the raw loss values of the original method and selecting a factor that brings our (scaled) loss to the same order of magnitude. Of course, this value can also be determined automatically. To illustrate this, we run experiments with DM as the baseline while keeping the curvature fixed at  -1. Here, the initial loss denotes the loss after multiplying by the scaling factor when the number of distribution‑matching iterations is zero (the original DM uses no scaling). Our experimental results are as follows:
>
> | IPC = 10 | $\lambda$ | Initial Loss | Accuracy (%) |
> |---------:|--:|-------------:|-------------:|
> |  DM | –  | 25 – 28 | 48.5 ± 0.6 |
> | DM + HDD | 10 | 3 – 4   | 50.3 ± 0.2 |
> | DM + HDD | 20 | 6 – 7   | 50.3 ± 0.3 |
> | DM + HDD | 40 | 12 – 14 | 50.2 ± 0.3 |
> | DM + HDD | 80 | 24 – 27 | 50.3 ± 0.2 |
>
> | IPC = 50 | $\lambda$ | Initial Loss | Accuracy (%) |
> |---------:|--:|-------------:|-------------:|
> |   DM | –  | 5 – 7  | 62.2 ± 0.5 |
> | DM + HDD | 40 | 5 – 7   | 63.0 ± 0.3 |
> | DM + HDD | 80  | 11 – 13 | 63.2 ± 0.4 |
> | DM + HDD | 120 | 17 – 19 | 63.3 ± 0.2 |
> | DM + HDD | 160 | 25 – 27 | 62.9 ± 0.2 |
>
> Because hyperbolic inter-centroid distances are inherently small, we multiply them by $\lambda$ to match the scale of the original DM loss. As long as $\lambda$ stays within a reasonable range, the accuracy remains virtually unchanged, confirming the **insensitivity of HDD to this scaling factor**.
>
> ---
>
> ### **Clarification 3 - Clarification of Notation and Equations**
>
> Regarding $K_i$ in line 265, what we originally meant was the temporal component of sample features embedded under a fixed curvature. We apologize for the resulting misunderstanding and will amend this in the revised manuscript. Equations (1) and (2) themselves are correct and follow a written format similar to DANCE [1], whereas the formulation you suggested aligns more with other distribution‑matching methods, so we will adopt the format in the revised version. Thank you also for your keen observation: we will rectify the notation issue and replace  $x_t$ and $x_s$ with $p_t$ and $p_s$.
>
> ---
>
> ### **Clarification 4 - Per-Sample Curvature**
>
> All samples were **consistently embedded in a hyperbolic space with identical curvature**. We apologize once again for the confusion regarding the definition of $K_i$  in line 265.
>
> ---
>
> ### **Clarification 5 - Can HDD Help Optimisation-Based DD?**
>
> HDD cannot boost optimization‑based methods on its own, but it can indirectly enhance their performance when used within a hybrid framework. As shown in Table  3 of the original paper, incorporating HDD into DANCE [1] yields performance improvements. We also provide results obtained by combining HDD with another hybrid approach, DSDM [2]. The table below summarises the results; **adding HDD yields consistent gains across all IPC settings on CIFAR-10**.
>
> | Method            | IPC | Accuracy (%) |
> |-------------------|----:|-------------:|
> | DSDM              |   1 | 43.8 ± 0.2   |
> | **DSDM + HDD**    | **1** | **46.4 ± 0.3** |
> | DSDM              |  10 | 65.8 ± 0.3   |
> | **DSDM + HDD**    | **10** | **66.6 ± 0.4** |
> | DSDM              |  50 | 75.8 ± 0.2   |
> | **DSDM + HDD**    | **50** | **76.0 ± 0.2** |
>
> ---
>
> ### **Clarification 6 - Evidence of Hierarchical Structure in Benchmarks**
>
> First, **the benchmark datasets we use also exhibit hierarchical structure**, which can be quantified by **Gromov δ‑hyperbolicity** [3]. Gromov δ‑hyperbolicity measures how close a metric space is to an ideal negatively curved space. Values near 0 indicate that the data are inherently more compatible with tree‑like or negatively curved geometry, whereas values approaching 1 suggest a stronger tendency toward Euclidean geometry. In feature space, common benchmark datasets have Gromov δ‑hyperbolicity of roughly as shown below:
>
> | Dataset      | δ-hyperbolicity |
> |--------------|----------------:|
> | CIFAR-10     | ≈ 0.25 |
> | CIFAR-100    | ≈ 0.23 |
> | TinyImageNet | ≈ 0.21 |
>
> These figures demonstrate that **these widely used benchmarks indeed lean toward negatively curved spaces and possess a certain degree of hierarchy**.
>
> ---
>
> We also visualized the labels of the five samples nearest to the origin and the five farthest from it among the 256 raw 'Horse' images (one batch) at the 10,000th training iteration of IDM on CIFAR-10. However, due to conference policy restrictions, we are unable to provide the images directly. You may inspect the corresponding images by downloading this publicly available dataset.  We apologize for any inconvenience this limitation may cause and appreciate your understanding.
>
> - **Nearest samples:** `data_batch_3_img6963`, `data_batch_4_img455`, `data_batch_4_img6399`, `data_batch_2_img4897`, `data_batch_4_img5922`
> - **Farthest samples:** `data_batch_1_img1394`, `data_batch_5_img897`, `data_batch_1_img2701`, `data_batch_3_img3910`, `data_batch_2_img6227`
>
> We observe that **samples farther from the origin are noticeably messier**, either showing cluttered scenes with both people and horses or only partial views such as a horse’s head, whereas those nearer to the origin are visually cleaner, free of distracting objects, and present the horse’s full outline.
>
> ---
>
> We hope that these detailed clarifications address all of your concerns. Thank you again for your valuable suggestions, which have helped us strengthen the manuscript.
>
> ---
>
> **References**
>
> [1] *DANCE: Dual-View Distribution Alignment for Dataset Condensation*
> [2] *Diversified Semantic Distribution Matching for Dataset Distillation*
> [3] *Hyperbolic Image Embeddings*

---

> > ### Comment · Reviewer_arwV · 2025-08-04
> >
> > Thank you for your rebuttal. The authors' response satisfactorily addresses my concerns. I appreciate the clarifications and additional evidence provided. I am willing to raise my score accordingly.

---

> > > ### Author Response · Authors · 2025-08-04
> > >
> > > Dear reviewer arwV, thank you for your continued support and insightful recommendations, which have been instrumental in improving our manuscript. In the final version, we will incorporate additional experiments as suggested. We sincerely appreciate your valuable feedback and look forward to further enhancing the completeness and impact of the paper.

---

### Official Review · Reviewer_RBVi · 2025-07-03

**Clarity:** 3
**Significance:** 2
**Originality:** 2
**Rating:** 3
**Confidence:** 4

**Summary:**

This study proposes Hyperbolic Dataset Distillation (HDD), a dataset distillation based on hyperbolic space, instead of Euclidean space-based dataset distillation. Since hyperbolic space naturally implies a hierarchical structure, we expect it to model the complex geometry and hierarchical relationship of datasets better than existing methods. As a specific methodology, this study proposes a distribution matching objective over hyperbolic space using Lorentz hyperbolic space. Experimental results show the effectiveness of HDD.

**Questions:**

1. The experimental results (Table 1) show that adding HDDs improves performance, but I'm curious about the rationale for this.

2. In this manuscript, it is mentioned that the closer to the centroid, the better the representation of category prototypes because it encodes the overall geometric structure of the dataset. However, it is difficult to accept the claim because it is difficult to judge which image is higher-level or lower-level when looking at the visualization of distilled images in the Appendix. I would like to know what hyperbolic space the distilled images should be visualized in and the interpretation of it.

3. I'm also curious about the results of HDD on a benchmark dataset with a hierarchy structure.

4. If a hierarchy structure exists in the optimized synthetic dataset, it is able to prune along the hierarchy after training. If this direction is correct, I would be interested to see a comparison with MDC [1], as well.

[1] Multisize Dataset Condensation

**Ethical Concerns:**

["NO or VERY MINOR ethics concerns only"]

**Final Justification:**

This paper introduces hyperbolic space instead of Euclidean space to the field of dataset distillation for the first time and proposes a dataset distillation objective through Riemannian mean matching. These ideas have been experimentally proven to show consistent performance improvements despite being intuitive and simple methodologies.

However, there is a lack of logical and solid grounds for introducing hyperbolic space to dataset distillation. Additionally, the analysis of the reasons for the performance improvement of the proposed methodology compared to previous studies has not been logically proven. Although these issues were discussed during the rebuttal period, clear and sufficient answers were not provided. In the absence of such evidence, I believe that the proposed simple methodology is merely a well-combined version of already widely known concepts.

Therefore, I propose a borderline rejection of the paper, as it requires further refinement in its core aspects.

**Limitations:**

See weaknesses and questions

**Paper Formatting Concerns:**

No paper formatting concerns

**Quality:**

3

**Strengths And Weaknesses:**

**Strengths**
1. This manuscript is well written and easy to read.

2. To my knowledge, this is the first study in the field of dataset distillation to discuss optimization in hyperbolic space rather than Euclidean space, which is interesting and significant because it brings new perspectives and possibilities to the field of dataset distillation.

3. This study proposes a reasonable and logical methodology using hyperbolic geometry. It is also easy to apply because HDD can be implemented with the materials of prior research without additional materials.

**Weaknesses**
1. My major concern is the originality and novelty of the work. As the authors note in Related works, the direction of introducing hyperbolic spaces into machine learning has been well studied. The proposed HDD is a methodology that replaces the Euclidean nature of vanilla DM with a hyperbolic nature. This work appears to be a combination of existing techniques, applying hyperbolic geometry to dataset distillation. Therefore, I think that this study lacks originality and novelty because HDD has not been modified for dataset distillation.

2. Lack of a clear rationale for applying hyperbolic space to dataset distillation. I was expecting to read about what problems Euclidean space DM has and how introducing hyperbolic space can solve them, but the manuscript only has statements to support this, and no theoretical or experimental evidence (only performance comparison).

3. Although hyperbolic space has the advantage of an inherent hierarchical structure, there are no experimental results for benchmark datasets that contain a hierarchical structure. This makes it difficult to verify that synthetic datasets created with HDDs actually contain a hierarchical structure well.

---

> ### Author Rebuttal · Authors · 2025-07-31
>
> ## Response to Reviewer RBVi
>
> We sincerely thank you for your insightful review and for acknowledging the clarity of our manuscript and the potential of introducing hyperbolic geometry into dataset distillation.  Below, we provide detailed responses to your concerns.
>
> ---
>
> ### **Clarification 1 – Originality**
>
> To the best of our knowledge, this is **the first study to integrate hyperbolic geometry into dataset distillation and to tailor the distribution‑matching objective explicitly to the curvature of the space**. We not only introduce a new distillation objective, matching the Fréchet means of two data distributions in hyperbolic space, but also **systematically examine how samples at different hierarchical levels contribute to the loss**, and we propose a hierarchy‑aware pruning strategy. Previous hyperbolic research has focused mainly on graph networks, metric learning, or generative models; in contrast, our framework targets the specific pain point in dataset distillation where hierarchical alignment and structure are often overlooked, as differentiated in our related work overview. In summary, we inject a hyperbolic hierarchical inductive bias into the distillation process for the first time, **without adding any extra training tricks, yet still achieve notable accuracy gains across multiple datasets** compared with baseline methods. This simplicity makes our approach easy to reproduce and extend. Further, we list some possible optimization directions: the framework can be generalized to (a) pseudo-Riemannian manifolds, (b) information-theoretic metrics in hyperbolic space, and (c) higher-order moment matching on hyperbolic manifolds, opening new avenues for more powerful and versatile dataset-distillation methods.
>
> ---
>
> ### **Clarification 2 – Motivation for Hyperbolic Space**
>
> **Limitations of Euclidean discrepancy measures.**
> Euclidean MSE / MMD treats samples as i.i.d. points, weighting high-level (fine-detail or noisy) samples the same as low-level (prototype) samples and thereby diluting distillation effects.
>
> **Why negative curvature?**
> Hyperbolic space ($K < 0$) expands exponentially in volume, embedding tree structures continuously. By matching centroids in this space, we align the hierarchical structure between real and synthetic data. Samples at larger geodesic distance naturally contribute less, **amplifying prototypes while suppressing noise**.
>
> **Experimental evidence.**
> Our pruning study (Table 2 in the paper) confirms that removing samples farther from the hyperbolic origin has little impact, indicating that those points indeed contain less critical information.
>
> ---
>
> ### **Clarification 3 – Hierarchical Properties of Benchmark Datasets**
>
> **The benchmark datasets we use also exhibit hierarchical structure**, which can be quantified by **Gromov δ‑hyperbolicity** [1]. Gromov δ‑hyperbolicity measures how close a metric space is to an ideal negatively curved space. Values near 0 indicate that the data are inherently more compatible with tree‑like or negatively curved geometry, whereas values approaching 1 suggest a stronger tendency toward Euclidean geometry. In feature space, common benchmark datasets have Gromov δ‑hyperbolicity of roughly as shown below:
>
> | Dataset      | δ-hyperbolicity |
> |--------------|----------------:|
> | CIFAR-10     | ≈ 0.25 |
> | CIFAR-100    | ≈ 0.23 |
> | TinyImageNet | ≈ 0.21 |
>
> These figures demonstrate that **these widely used benchmarks indeed lean toward negatively curved spaces and possess a certain degree of hierarchy**.
>
> ---
>
> ### **Clarification 4 – Rationale for Performance Improvements**
>
> As we explained in **Clarification 2**, our choice is grounded in two main considerations:
>
> (a) By aligning centroids to match the hierarchical structure of the data, we minimize the overall distributional gap between synthetic and real samples. As Figure  4 shows, the synthetic samples closely trace the distribution of the originals.
>
> (b) The hierarchical weighting of centroids in hyperbolic space encourages the synthetic samples toward class prototypes while suppressing boundary noise.
>
> ---
>
> ### **Clarification 5 – Visualisation and Interpretation**
>
> Due to conference policy, images cannot be uploaded. We therefore list IDs of the five CIFAR-10 *Horse* images **nearest to** and **farthest from** the hyperbolic origin after 10,000 IDM iterations:
>
> - **Nearest samples:** `data_batch_3_img6963`, `data_batch_4_img455`, `data_batch_4_img6399`, `data_batch_2_img4897`, `data_batch_4_img5922`
> - **Farthest samples:** `data_batch_1_img1394`, `data_batch_5_img897`, `data_batch_1_img2701`, `data_batch_3_img3910`, `data_batch_2_img6227`
>
> When viewed locally, **near-origin samples are visually cleaner and centred**, whereas far-origin samples are cluttered or partially occluded, supporting our interpretation of hyperbolic radius as a hierarchy indicator.
>
> ---
>
> ### **Clarification 6 – Benchmark Dataset with a Hierarchy Structure**
>
> As we explained in **Clarification 3**, we have confirmed that the benchmark dataset we used also exhibits a hierarchical structure. In the future, we plan to extend HDD to other data modalities, such as language, with even richer hierarchical structures.
>
> ---
>
> ### **Clarification 7 – Comparison with Multisize Dataset Condensation**
>
> Multisize Dataset Condensation (MDC) collapses the N individual condensation runs that would normally be required for different target set sizes into a single process. It does so by adding an adaptive subset loss on top of the standard gradient‑matching loss, allowing the full synthetic set and any of its subsets to converge toward the real data simultaneously and eliminating the “subset performance collapse.” In other words, **MDC permits informational overlap between a subset and the complete set**. Our method, in contrast, **attaches explicit hierarchical information to every synthetic sample**. Trimming the synthetic dataset, therefore, means deliberately discarding part of that hierarchy, which causes a pronounced performance drop. The results of the experiment we conducted are as follows：
>
> | Scenario (CIFAR-10)                  | Accuracy (%) |
> |--------------------------------------|--------------|
> | IPC 1                             | 28.7 ± 0.2 |
> | IPC 10 → IPC 1             | 21.6 ± 0.8 |
> | IPC 50 → IPC 1             | 17.0 ± 0.6 |
>
> We observed that **the larger the IPC in the original set, the less information remains after trimming**. Specifically, cutting an IPC‑50 set down to IPC‑1 performs worse than cutting an IPC‑10 set down to IPC‑1, while a native IPC‑1 set, with no trimming at all, achieves the best results by a wide margin.
>
> ---
>
> We hope that these detailed clarifications address all of your concerns. Thank you again for your valuable suggestions, which have helped us strengthen the manuscript.
>
> ---
>
> **Reference**
>
> [1] *Hyperbolic Image Embeddings*.

---

> > ### Comment · Reviewer_RBVi · 2025-08-04
> >
> > I appreciate the authors’ effort during the rebuttal process. Most of my concerns have been addressed. However, there still remains a question regarding the rationale behind why hyperbolic space contributes to the improvement of dataset distillation performance. Let us define the true data distribution $P$ and synthetic data distribution $Q$ in Euclidean space, respectively. Similarly, we define the true and synthetic data distributions in hyperbolic space as $P_\psi$ and $Q_\psi$. The original DM approach performs risk approximation $E_{P}[\cdot] \approx E_{Q}[\cdot]$ through distribution matching $P \approx Q$, ultimately aiming for parameter approximation.
> >
> > I believe that a similar line of reasoning is necessary in the proposed methodology, and the following questions could help justify the introduction of hyperbolic space:
> >
> > 1. Since Riemannian mean matching appears to be a form of first-order moment matching, I wonder whether hyperbolic distribution matching ($P_\psi \approx Q_\psi$) can be interpreted analogously to MMD (Maximum Mean Discrepancy).
> > 2. Is hyperbolic distribution matching ($P_\psi \approx Q_\psi$) connected to Euclidean distribution matching ($P \approx Q$) or the dataset distillation objective (Eq. 1)?
> > 3. If such a connection exists, does hyperbolic distribution matching guarantee a smaller distributional gap?
> >
> > Given that the authors emphasize distribution alignment throughout the paper, I believe such an interpretation from the distributional perspective should also be included. As it stands, the paper does not sufficiently support the claim that HDD ensures better distribution alignment. Figure 4, which the authors mention, only provides a qualitative analysis of applying HDD to the baseline. Quantitative analysis in comparison with the baseline is limited to performance metrics. Eq.19 in the main text shows that the proposed method relates to per-sample weighted Euclidean distance, which could serve as a good starting point.

---

> ### Author Response · Authors · 2025-08-04
>
> We appreciate your additional questions and are glad that most of the earlier concerns have been resolved. Below, we reply to the three new points.
>
> ---
>
> ### 1.
>
> The reviewer’s remark that “Riemannian-mean matching is essentially first-order moment matching” is exactly correct. Our method can indeed be viewed as **moment matching performed in hyperbolic space**. Conceptually, this parallels Euclidean moment-matching approaches, such as MMD; however, the implementation details and emphases differ.
>
> - **Common Goal**: Both our HDD and Euclidean MMD aim to minimise the distributional gap between the real and synthetic datasets by aligning key statistical properties.
>
> - **Key Difference**: Conventional DM aligns the Euclidean means (first-order moments), implicitly treating every sample as equally important. In contrast, **HDD matches the hyperbolic Riemannian mean (centroid)**. Because of negative curvature, the centroid is naturally biased toward samples closer to the origin (root-centric). Thus, by matching hyperbolic centroids, **HDD performs implicit hierarchy-aware re-weighting**, offering a more precise form of distribution alignment.
>
> ---
>
> ### 2.
>
> **Our pipeline is not detached from Euclidean space.** As detailed in the paper, we first map each sample into an Euclidean feature space via a pretrained encoder and then embed those features into hyperbolic space via the exponential map. Hence, the optimisation of $𝑃_{\psi} ≈ 𝑄_{\psi}$ is **based on Euclidean features**. The ultimate goal of dataset distillation (Eq. 1) is to minimise the performance gap between networks trained on real versus synthetic data under identical parameters. DM methods adopt a proxy objective: if real and synthetic data are well aligned in some feature space, then models trained on them should perform similarly. HDD follows the same logic but contends that performing the alignment in a non-Euclidean space that better captures intrinsic structure yields a stronger proxy, enabling the synthetic set to generalise better and thus more closely approach the original distillation objective. Empirically, this claim holds: e.g., on CIFAR-100 (IPC = 1), HDD improves accuracy over the IDM baseline by 3.2%.
>
> ---
>
> ### 3.
>
> Distribution matching seeks to shrink the gap between real and synthetic data in feature space. Although **the raw centroid distance under HDD is indeed numerically smaller**, this is **largely due to the geometry of hyperbolic space (distances near the origin are inherently shorter)**, which is why we introduce a **scaling factor**. Because hyperbolic and Euclidean distances are not metrically equivalent, cross-space magnitudes are hard to compare directly. In such cases, test accuracy serves as the most convincing evidence. Across nearly all baselines and datasets, **HDD yields consistent performance gains**, demonstrating that it achieves a closer distributional match to the original data.
>
> ---
>
> We hope these clarifications address the remaining questions and further justify the use of hyperbolic geometry in HDD. We will also include these clarifications in the revised manuscript. Thank you again for your helpful feedback.
>
> Looking forward to your reply!

---

> > ### Comment · Reviewer_RBVi · 2025-08-07
> >
> > Thank you for your additional response.
> >
> > The question of whether higher-order moment matching and/or hyperbolic distribution matching are guaranteed through Riemannian-mean matching, which is first-order moment matching in the hyperbolic space, remains unresolved. MMD theoretically guarantees distribution matching even if first-order moment matching is performed in Euclidean space for RKHS with high expressiveness. Without a well-defined theoretical basis in hyperbolic space, the authors' claim that distribution alignment can be achieved through Riemannian-mean matching is difficult to accept, currently.
> >
> > In their additional response, the authors mentioned that 1) Riemannian-mean matching is based on Euclidean features, but 2) direct comparison of distances between hyperbolic and Euclidean spaces is difficult, and 3) they confirmed a smaller distribution gap through HDD's high experimental performance. This claim, as mentioned in weakness 2 of my first review, still demonstrates the effectiveness of the proposed methodology solely through performance comparisons, without theoretical basis or other forms of experimental evidence. In my second response, I defined the notation and expected to see a theoretical connection between Euclidean distribution and hyperbolic distribution, but this was not confirmed in the authors' response. If theoretical analysis was difficult within the short rebuttal period, they could have presented experimental evidence other than performance comparisons, such as toy experiments. Since projecting the Riemannian mean onto Euclidean space is expected to yield a result different from the Euclidean mean, Riemannian mean matching is likely to have a different optimal solution than Euclidean mean matching. In this context, it cannot be definitively stated that Riemannian mean matching reduces the distribution gap more effectively than Euclidean mean matching.
> >
> > This paper introduces hyperbolic geometry, which has not been addressed in dataset distillation, for the first time and experimentally proves that the proposed methodology is effective despite its simplicity. However, logical and solid evidence for the utilization of hyperbolic geometry in dataset distillation and the resulting performance improvement needs to be supplemented. The authors additionally presented in the Response that the benchmark dataset has a hyperbolic structure, but this is insufficient as the core evidence that hyperbolic geometry contributes to improving dataset distillation performance. Therefore, I would like to maintain the current status.

---

> > > ### Author Response · Authors · 2025-08-08
> > >
> > > We thank the reviewer for highlighting the importance of clarifying the relationship between first-order and higher-order moment matching, as well as the connection between Euclidean and hyperbolic distribution matching in our formulation. Below, we provide additional conceptual and theoretical analysis, explaining how first-order matching behaves differently under Euclidean and hyperbolic geometries, and why this distinction matters for dataset distillation.
> > >
> > > ---
> > >
> > > **1. Definition and role of higher-order moment matching**
> > >
> > > In distribution matching, the *moments* of a distribution capture progressively richer statistical characteristics:
> > >
> > > * First-order moment: mean (location)
> > > * Second-order moment: covariance (scale, correlation)
> > > * Higher-order moments: skewness, kurtosis, and beyond (shape, tail behavior)
> > >
> > > In Euclidean space, MMD with a **characteristic kernel** (e.g., Gaussian RBF) implicitly matches *all* moments. This is why, in a sufficiently expressive RKHS, aligning only the first-order moment vector in that space guarantees alignment of the full distributions.
> > >
> > > ---
> > >
> > > **2. Difference between Euclidean MMD and HDD**
> > >
> > > Most dataset-distillation baselines (DM, IDM, DANCE, DSDM, IID) operate with **linear kernels** in Euclidean space. In this case, MMD degenerates to first-order moment matching — no higher-order terms are captured. HDD replaces this Euclidean centroid matching with **Riemannian mean matching in hyperbolic space**, introducing a *geometry-aware weighting* effect: points near the origin have disproportionately higher influence on the mean, while peripheral points are downweighted.
> > >
> > > This difference in weighting alters the optimal solution. In Sturm’s framework \[1], the barycenter map in nonpositively curved spaces satisfies:
> > >
> > > $$
> > > d_\mathbb{H}(\mu_\mathbb{H}(P), \mu_\mathbb{H}(Q)) \leq W_2^\mathbb{H}(P, Q),
> > > $$
> > >
> > > ensuring that barycenter matching is **non-expansive** and preserves global geometric relationships. While this does not provide a full higher-order guarantee, it bounds the centroid distance in terms of the underlying Wasserstein distance, which itself aggregates differences in *all* moments.
> > >
> > > Regarding the metric issue between hyperbolic and Euclidean spaces: our method performs distribution matching **within** hyperbolic space, minimising the distribution discrepancy in that geometry. Euclidean distribution matching corresponds to minimising discrepancy in Euclidean space. Mapping either distribution into the other inevitably introduces distortion, making direct cross-space comparison theoretically ill-posed. In such cases, downstream performance is the most practical and reliable proxy for distribution alignment, consistent with the original motivation of distribution-matching tasks.
> > >
> > > We agree that Riemannian mean matching and Euclidean mean matching have different optimal solutions. This difference is precisely where HDD’s advantage lies: the curvature-induced weighting biases the optimisation toward more prototypical, hierarchically central samples, leading to consistent performance gains. If both losses had the same optimal solution, the distinction between the methods would be moot.
> > >
> > > ---
> > >
> > > **3. On theoretical guarantees for higher-order matching in hyperbolic space**
> > >
> > > Unlike Euclidean RKHS theory, the study of characteristic kernels and moment completeness in hyperbolic manifolds is still nascent. Proving an analogue of the Euclidean MMD guarantee for general higher-order moments in hyperbolic space is non-trivial, because:
> > >
> > > 1. The tangent space metric distorts higher-order interactions when mapped back to the manifold.
> > > 2. Kernel constructions on manifolds must respect geodesic distances, curvature, and volume growth.
> > >
> > > For these reasons, our current work focuses on the first-order term, where the theoretical behaviour is clear, the computation is stable, and the inductive bias (prototype emphasis) is beneficial in low-IPC regimes.
> > >
> > > ---
> > >
> > > **4. Future extension**
> > >
> > > We agree that incorporating higher-order moment matching directly in hyperbolic space is a promising direction. For example:
> > >
> > > * **Second-order**: Match covariance matrices in the tangent space at the hyperbolic mean, weighted by curvature-adjusted volume elements.
> > > * **Kernelised approach**: Develop characteristic kernels on hyperbolic manifolds to extend the RKHS framework and recover full moment-matching guarantees.
> > >
> > > ---
> > >
> > > \[1] K.-T. Sturm. *Probability Measures on Metric Spaces of Nonpositive Curvature*. Contemporary Mathematics, 338, 2003.

---

### Official Review · Reviewer_gver · 2025-07-03

**Clarity:** 3
**Significance:** 3
**Originality:** 2
**Rating:** 4
**Confidence:** 4

**Summary:**

This paper introduces Hyperbolic Dataset Distillation (HDD), a drop-in modification to existing distribution-matching methods (DM, IDM, Dance) that embeds real and synthetic features into Lorentz hyperbolic space and minimizes the geodesic distance between their centroids. The key insight is that hyperbolic geometry inherently emphasizes low-level “prototype” samples, which helps retain class semantics and hierarchical structure during distillation. The authors demonstrate that HDD consistently improves performance across five datasets (Fashion-MNIST, CIFAR-10, CIFAR-100, SVHN, Tiny-ImageNet) for a variety of IPC (images per class) settings and enables “hierarchical pruning,” where only 20% of the real data is used to distill without hurting final accuracy.

**Questions:**

1. How sensitive is HDD to the curvature K? Are there stability issues for very negative values?

2. Have you tried using more sophisticated centroids as opposed to just tangent-space approximation of the Fréchet mean? In fact, due to some recent work on differentiating through the Fréchet mean [a], you may actually be able to perform the operation with the true mean (although this will likely be considerably slower).

### References

[a] Differentiating through the Fréchet Mean. https://arxiv.org/pdf/2003.00335

**Ethical Concerns:**

["NO or VERY MINOR ethics concerns only"]

**Final Justification:**

Fundamentally, the paper is a clean and reasonably well-executed work that introduces a simple yet effective modification to DM-style dataset distillation. After the rebuttal, I still believe the originality is fairly limited, but am happy that the authors have presented runtime/memory comparisons in their rebuttal and would recommend these be included in the final paper. I also very much like the addition of a comparison between the tangent space mean and the Fréchet mean (I also highly recommend the authors include this in the final version). Overall, my opinion is that rebuttal has improved the paper, but fundamentally I still maintain my weak accept rating.

**Limitations:**

Yes

**Paper Formatting Concerns:**

I have no major formatting concerns for this paper.

**Quality:**

3

**Strengths And Weaknesses:**

## Strengths

1. The main idea of doing centroid matching in Lorentz space is simple, intuitive, and reasonably novel in the distillation setting. The use of hyperbolic geometry biases the synthetic data toward prototypical examples and aligns well with the distribution-matching objective.

2. The improvements are consistent across multiple datasets and baselines. For instance, adding HDD to IDM yields a 4% absolute gain on CIFAR-10 at IPC=10 (from 57.3% to 61.3%). Similar trends hold across other datasets, including CIFAR-100 and Tiny-ImageNet. Standard deviation are given and in most cases the proposed method yields a statistically significant result.

3. The hierarchical pruning results are strong. The fact that the authors can discard 80% of the real dataset and still get the same final accuracy is surprising and interesting. The pruning also appears to stabilize training.

## Weaknesses

1. The paper lacks any ablation over curvature ($K$) or the weight $\lambda$ on the centroid loss. These hyperparameters likely matter quite a bit, especially given the numerical instability of hyperbolic operations, and should be studied.

2. No runtime or memory comparison is given. Log/exp maps and acosh are not cheap, and it’s unclear whether this adds significant overhead over the Euclidean baseline.

3. Although the idea is new in the context of dataset distillation, it’s not fundamentally new in general. Centroid matching is well-known, and using a different geometry for it is not that far from existing work.

4. The scope is fairly limited: experiments top out at Tiny-ImageNet (64×64 resolution), and it’s unclear whether the method scales to larger datasets like ImageNet-1k or non-image domains.

## Verdict

This is a clean and reasonably well-executed paper that introduces a simple yet effective modification to DM-style dataset distillation. While the originality is limited and runtime/memory comparisons are missing, the empirical results are solid and the implementation is relatively straightforward. I recommend a weak accept rating.

---

> ### Author Rebuttal · Authors · 2025-07-31
>
> ## Response to Reviewer gver
>
> We sincerely appreciate your thoughtful review and constructive comments on our submission. Below, we provide detailed answers to each of your questions and comments.
>
> ---
>
> ### **Clarification 1 – Ablation on Curvature $K$ and Loss-Weight $\lambda$**
>
> We conducted an ablation study on different curvature values $K$ within the DM framework on CIFAR-10; the results are presented below:
>
> ---
>
> | IPC   | Method       | $\|K\|$ = 1/3       | 0.5             | 1               | 2               | 5               |
> |-------|--------------|----------------|------------------|------------------|------------------|------------------|
> | 1     | DM           | 26.4 ± 0.3     | 26.4 ± 0.3      | 26.4 ± 0.3      | 26.4 ± 0.3      | 26.4 ± 0.3      |
> |       | DM + HDD     | 27.0 ± 0.2     | 28.8 ± 0.3      | 28.7 ± 0.2      | 27.6 ± 0.2      | 28.6 ± 0.2      |
> | 10    | DM           | 48.5 ± 0.6     | 48.5 ± 0.6      | 48.5 ± 0.6      | 48.5 ± 0.6      | 48.5 ± 0.6      |
> |       | DM + HDD     | 49.6 ± 0.3     | 49.9 ± 0.1      | 50.3 ± 0.3      | 50.1 ± 0.1      | 50.0 ± 0.2      |
> | 50    | DM           | 62.2 ± 0.5     | 62.2 ± 0.5      | 62.2 ± 0.5      | 62.2 ± 0.5      | 62.2 ± 0.5      |
> |       | DM + HDD     | 63.0 ± 0.3     | 63.1 ± 0.1      | 63.2 ± 0.4      | 63.1 ± 0.2      | 62.7 ± 0.1      |
>
> Although $\|K\|$ slightly affects the final accuracy, **the variation is modest, and HDD consistently surpasses the Euclidean baseline**.
>
> ---
>
> Unlike the curvature hyperparameter, the inter‑centroid distances in hyperbolic space are extremely small because of its unique geometry, so we introduce a scaling factor. **As long as this factor stays within a reasonable range, it exerts little influence on the final results.** We choose its value empirically: by inspecting the raw loss values of the original method and selecting a factor that brings our (scaled) loss to the same order of magnitude. Of course, this value can also be determined automatically. To illustrate this, we run experiments with DM as the baseline while keeping the curvature fixed at  -1. Here, the initial loss denotes the loss after multiplying by the scaling factor when the number of distribution‑matching iterations is zero (the original DM uses no scaling). Our experimental results are as follows:
>
> | IPC = 10 | $\lambda$ | Initial Loss | Accuracy (%) |
> |---------:|--:|-------------:|-------------:|
> |  DM | –  | 25 – 28 | 48.5 ± 0.6 |
> | DM + HDD | 10 | 3 – 4   | 50.3 ± 0.2 |
> | DM + HDD | 20 | 6 – 7   | 50.3 ± 0.3 |
> | DM + HDD | 40 | 12 – 14 | 50.2 ± 0.3 |
> | DM + HDD | 80 | 24 – 27 | 50.3 ± 0.2 |
>
> | IPC = 50 | $\lambda$ | Initial Loss | Accuracy (%) |
> |---------:|--:|-------------:|-------------:|
> |   DM | –  | 5 – 7  | 62.2 ± 0.5 |
> | DM + HDD | 40 | 5 – 7   | 63.0 ± 0.3 |
> | DM + HDD | 80  | 11 – 13 | 63.2 ± 0.4 |
> | DM + HDD | 120 | 17 – 19 | 63.3 ± 0.2 |
> | DM + HDD | 160 | 25 – 27 | 62.9 ± 0.2 |
>
> Because hyperbolic inter-centroid distances are inherently small, we multiply them by $\lambda$ to match the scale of the original DM loss. As long as $\lambda$ stays within a reasonable range, the accuracy remains virtually unchanged, confirming the **insensitivity of HDD to this scaling factor**.
>
> ---
>
> ### **Clarification 2 – Runtime and Memory Overhead**
>
> We benchmarked 100 training iterations on a single RTX 4090 GPU with batch size = 256 on CIFAR-10.
>
> | IPC | DM Runtime | DM + HDD Runtime | DM Memory | DM + HDD Memory |
> |-----|------------|------------------|-----------|-----------------|
> | 1   | 4.9 s      | 6.7 s            | 3522 MiB  | 3522 MiB        |
> | 10  | 5.0 s      | 6.8 s            | 3626 MiB  | 3632 MiB        |
> | 50  | 5.4 s      | 7.1 s            | 3888 MiB  | 3922 MiB        |
>
> **Memory usage for the two methods is nearly identical**, while HDD requires more runtime, but the overhead remains within an acceptable range.
>
> ---
>
> ### **Clarification 3 – Originality**
>
> To the best of our knowledge, this is **the first work to integrate hyperbolic geometry into dataset distillation** and to **tailor the distribution-matching objective explicitly to the space’s curvature**. We further observe that the synthetic samples naturally mimic the hierarchical structure of the real data and gravitate toward the class centroid. Exploiting this property, we prune redundant real samples without any loss of accuracy. We have deliberately kept the method simple, **no additional training tricks are introduced**, but we still **achieve notable accuracy gains over strong baselines on multiple datasets**. This simplicity enhances both reproducibility and extensibility. Further, we list some possible optimization directions: the framework can be generalized to (a) pseudo-Riemannian manifolds, (b) information-theoretic metrics in hyperbolic space, and (c) higher-order moment matching on hyperbolic manifolds, opening new avenues for more powerful and versatile dataset-distillation methods.
>
> ---
>
> ### **Clarification 4 – Scalability to Larger Datasets and Non-Image Domains**
>
> We have applied HDD to an ImageNet subset, ImageWoof, with high resolution (224 × 224), and verified the effectiveness of our method (Table 3). The results indicate that HDD can scale beyond tiny images, and we are planning to extend experiments to other modalities like text and speech.
>
> ---
>
> ### **Clarification 5 – Sensitivity and Stability at Extreme Curvatures**
>
> We conducted experiments on the extreme values of $\|K\|$​​ (including both extremely high curvature and extremely low curvature), and the results are as follows:
>
> | IPC   | Method       | No Curvature     | $\|K\|$ = 0.01 (Low Curvature) | $\|K\|$ = 100 (High Curvature) |
> |-------|--------------|------------------|----------------------------|----------------------------|
> | 1     | DM           | 26.4 ± 0.3       | -                          | -                          |
> | 1     | DM + HDD     | -                | 26.6 ± 0.2                 | 27.8 ± 0.2                 |
> | 10    | DM           | 48.5 ± 0.6       | -                          | -                          |
> | 10    | DM + HDD     | -                | 48.6 ± 0.3                 | 49.9 ± 0.2                 |
> | 50    | DM           | 62.2 ± 0.5       | -                          | -                          |
> | 50    | DM + HDD     | -                | 62.5 ± 0.3                 | 63.1 ± 0.1                 |
>
> We observe that when the curvature $\|K\|$ is very small, the performance gap between HDD and the original DM shrinks, which aligns with the theoretical expectation that hyperbolic space degenerates into Euclidean space as the curvature approaches zero. Conversely, when $\|K\|$ becomes very large, our method remains effective.
>
> ---
>
> ### **Clarification 6 – Exact vs. Tangent-Space Fréchet Mean**
>
> The vast majority of hyperbolic geometry methods in machine learning employ tangent space (or other approximate) centroids, as they are numerically stable and have consistently proven effective in practice. This is particularly true for dataset distillation frameworks, where centroids are invoked thousands of times per training run, making runtime efficiency critically important. We conducted experiments using the exact centroid computation method from the paper you provided. For the differential approximation, we perform 20 iterations to compute its centroid. In both centroid‑computation methods, the curvature is fixed at  -1. The runtime (for 100 iterations) and results on CIFAR-10 are as follows:
>
> | IPC | DM (Accuracy, Time)     | HDD (Tangent)             | HDD (Exact FM)           |
> |-----|--------------------------|----------------------------|---------------------------|
> | 1   | 26.4 ± 0.3, 4.9 s        | 28.7 ± 0.2, 6.7 s         | 28.4 ± 0.4, 8.6 s        |
> | 10  | 48.5 ± 0.6, 5.0 s        | 50.3 ± 0.3, 6.8 s         | 50.4 ± 0.2, 9.5 s        |
> | 50  | 62.2 ± 0.5, 5.4 s        | 63.2 ± 0.4, 7.1 s         | 63.3 ± 0.3, 9.8 s        |
>
> It can be seen that the two approaches produce nearly identical results, yet the computational burden rises markedly. Therefore, we consider the tangent‑space approximation to be the more cost‑effective option. We will add a note on this point in the revised version. Thank you for your valuable suggestion!
>
> ---
>
> We hope that these detailed clarifications address all of your concerns. Thank you again for your valuable suggestions, which have helped us strengthen the manuscript.

---

> > ### Comment · Reviewer_gver · 2025-08-06
> > **Response**
> >
> > Thank you for the rebuttal; I appreciate the new experiments and clarifications, which addressed most of my concerns. The curvature and scaling ablations are helpful; it’s good to see that performance is relatively stable across a wide range of values, including extreme curvature settings.
> >
> > I still believe the core idea is more incremental than it is transformative, but the given emphasis on simplicity and extensibility is reasonable.
> >
> > The additional test with exact Fréchet means was quite interesting; I find it reassuring that the tangent-space approximation remains competitive given the lower computational cost.
> >
> > My opinion is that the rebuttal strengthens the submission; my evaluation remains the same (weak accept), but I’m more confident in the authors' empirical claims and the robustness of the method.

---

> > > ### Author Response · Authors · 2025-08-06
> > >
> > > Dear reviewer gver, we sincerely thank you for your thoughtful follow-up and for taking the time to review our additional experiments. We are glad the ablations and the exact‐Fréchet-mean test addressed your remaining concerns. We will incorporate these new results and related implementation details into the revised version to improve reproducibility. We appreciate your increased confidence in the robustness of our method.

---

### Official Review · Reviewer_9awn · 2025-07-07

**Clarity:** 3
**Significance:** 3
**Originality:** 4
**Rating:** 5
**Confidence:** 4

**Summary:**

Dataset distillation compresses a large dataset into a small set of synthetic examples that achieve performance comparable to training on the full original data. In dataset distillation methods, distribution matching algorithms align feature distributions between real and synthetic dataset. This paper aims to leverage the hierarchical structure inherent in dataset samples, and introduce a novel hyperbolic data distillation (HDD) method. The objective in distribution matching is reformulated as minimizing the Lorentzian hyperbolic distance between the Riemannian means of the original and synthetic datasets. Experiments show that the proposed HDD method is easy to add on top of two existing method across 5 different datasets leading to better performance improvements.

**Questions:**

Main questions are listed in weaknesses. To summarize, I suggest the authors to clarify how the cross entropy loss of IDM and DANCE is added to HDD, whether as hyperbolic or euclidean? I also suggest the authors to visualize the samples of synthetic and real datasets that are closer to the origin and therefore are more _important_ for distribution matching. Finally, I want to know the authors' perspective on how a class-wise constraint can be added to HDD?

**Minor**:
- What is IPC? IPC of synthetic dataset is referred to multiple times in the paper but is not explained.
- duplicate citations ([23],[24])

**Ethical Concerns:**

["NO or VERY MINOR ethics concerns only"]

**Final Justification:**

The authors have addressed my concerns, so I keep my original rating. I suggest the authors to add proposed changes to the final version

**Limitations:**

The authors discuss limitations adequately.

**Paper Formatting Concerns:**

No formatting concerns

**Quality:**

3

**Strengths And Weaknesses:**

## Strengths

**Novelty:** To my knowledge, this is the first work to explore dataset distillation through the lens of hyperbolic geometry. The formulation of distribution matching loss in hyperbolic space is simple but sound.

**Interesting observations:** The observation that samples closer to origin contribute more strongly to the Fréchet mean of the dataset is interesting. The authors equate this to reducing outlier contribution (line 213-214). This is also validated in Table 2 where removing 80% samples exhibiting highest curvature does not impact the performance on CIFAR-10 dataset.

**Empirical Effectiveness:** The authors mention that HDD can be *plugged* into existing distribution matching methods and achieves meaningful gains over over using the methods alone. The experiments show results across distribution matching methods like DM[62], IDM[64] and hybrid methods like DANCE[58] (alternates between cross entropy and distribution matching).

## Weaknesses

**How can HDD augment existing distribution matching methods?**

- Firstly, It is straightforward to understand how HDD loss can replace maximum mean discrepancy loss (MMD) the DM[62] paper. However, there are some missing implementation details with IDM[64] and DANCE[58]. Since both methods have an additional cross entropy loss, is the cross entropy loss still in euclidean space or in hyperbolic space? What is reason for choosing one or the other?
- The paper only shows two distribution matching methods, the original DM[62] and IDM[64]. Is this generalizable to other distribution matching methods like CAFE [50], M3D[59], IID[12]? Is it as simple as replacing the MMD loss with HDD loss or does it require additional tuning?

**Visualizations**: Which samples contribute the most/least in synthetic dataset? I suggest the authors to also visualize such samples. Do samples closer to origin look more prototypical than samples farther away suggested in Figure 1.

**Class wise contributions:** Figure 4 shows that samples clustering is quite disperse. This will affect class-wise separability. How does class-aware regularization (similar to IDM or IID) look like in hyperbolic space? While this doesn't affect the novelty introduced in the paper, I'm curious how a hyperbolic equivalent of class centralization constraint (compact clustering in IID) look like and how would it impact how samples look like in the space?

---

> ### Author Rebuttal · Authors · 2025-07-31
>
> ## Response to Reviewer 9awn
>
> We sincerely thank you for the thoughtful review and for highlighting the novelty and empirical value of our approach. Below, we provide detailed answers to each of your questions and comments.
>
> ---
>
> ### **Clarification 1 – About the Cross-Entropy Term in IDM and DANCE**
>
> In our current implementation, **the cross-entropy (CE) loss remains in Euclidean space**. The logits produced by standard convolutional networks live naturally in Euclidean tensor space, and the CE loss is well defined there. By leaving this component unchanged, **we guarantee full compatibility with existing dataset-distillation baselines** and avoid introducing additional confounding factors. Although a hyperbolic softmax layer is conceptually possible, it would require redesigning the classifier head and decision boundaries, which we consider outside the scope of this study.
>
> ---
>
> ### **Clarification 2 – Generalisability to Other Distribution-Matching Methods**
>
> For methods such as CAFE that align layer-wise activations, the objective differs substantially from our global centroid-matching formulation; an effective combination would therefore demand a dedicated re-design of the loss. M3D focuses on higher-order moment matching and can, in principle, benefit from our hyperbolic formulation as well, but the resulting optimisation landscape would become considerably more complex.
>
> To substantiate the claim of broad applicability, **we have already integrated HDD into DSDM [1]**, which is a hybrid algorithm that alternates between CE and distribution matching. The table below summarises the results; **adding HDD yields consistent gains across all “images-per-class” (IPC) settings on CIFAR-10**.
>
> | Method            | IPC | Accuracy (%) |
> |-------------------|----:|-------------:|
> | DSDM              |   1 | 43.8 ± 0.2   |
> | **DSDM + HDD**    | **1** | **46.4 ± 0.3** |
> | DSDM              |  10 | 65.8 ± 0.3   |
> | **DSDM + HDD**    | **10** | **66.6 ± 0.4** |
> | DSDM              |  50 | 75.8 ± 0.2   |
> | **DSDM + HDD**    | **50** | **76.0 ± 0.2** |
>
> ---
>
> We also conducted the IPC-10 experiment on CIFAR-10 using IID (DM version). We only replaced the MMD component, setting curvature $K= -1$ and the scaling factor $\lambda$ to 20. Our experimental results are as follows:
>
> ---
>
> | Method        | Accuracy (%) |
> |---------------|-------------:|
> | IID (DM)      | 55.1 ± 0.1 |
> | **IID + HDD** | **55.8 ± 0.2** |
>
> ---
>
> Due to time constraints, experiments that substitute the MMD term with HDD in IID for additional settings are still running, and we will report the corresponding results in the revised manuscript.
>
> ---
>
> ### **Clarification 3 – Prototypicality of Samples Near the Hyperbolic Origin**
>
> Because conference policy prevents us from embedding raw CIFAR-10 images directly in the rebuttal, we instead list the identifiers of **the five samples closest to the origin** and **the five farthest** after 10,000 optimisation steps of IDM (all from the *horse* class).
>
> - **Nearest samples:** `data_batch_3_img6963`, `data_batch_4_img455`, `data_batch_4_img6399`, `data_batch_2_img4897`, `data_batch_4_img5922`
> - **Farthest samples:** `data_batch_1_img1394`, `data_batch_5_img897`, `data_batch_1_img2701`, `data_batch_3_img3910`, `data_batch_2_img6227`
>
> When we visualise these images locally, **near-origin samples are centred and uncluttered**, whereas far-origin samples are partially occluded, truncated, or mixed with background objects. This qualitative difference supports our claim that hyperbolic geometry naturally emphasises prototypical examples.
>
> ---
>
> ### **Clarification 4 – Adding Class-Wise Regularisation in Hyperbolic Space**
>
> The visualization results for IDM are already provided in our paper, specifically in the two panels on the right side of Figure 4. In addition, we are also visualizing IID. However, due to conference policy restrictions, we are unable to directly include the corresponding images. We will include the visualization results in the revised version.
>
> We agree that encouraging intra-class coherence without compromising the global hierarchy is an important direction. We have revisited the IID approach: when IID introduces class‑centric constraints, directly applying it in hyperbolic space may be unsuitable, because **those constraints can compromise the dataset’s hierarchical structure**. So we propose a different angle: **the overall hyperbolic distance can be decomposed into radial and angular components**. We can strip out the radial component—since forcibly shortening it might damage the hierarchy—while retaining the angular component to impose intra‑class constraints. Next, we provide the specific definition of angular loss for two points. Let a Lorentz-model point be written as $\(x=(x_0,\mathbf{x})\)$ with $\(x_0>0\)$ and $\(-x_0^{2}+\|\mathbf{x}\|^{2}=-1\)$. For two points $\(x\)$ and $\(y\)$ we extract their spatial parts and normalise them,
>
> $$
> \hat n_x = \frac{\mathbf{x}}{\|\mathbf{x}\|}, \quad
> \hat n_y = \frac{\mathbf{y}}{\|\mathbf{y}\|},
> $$
>
> then compute the directional cosine $\(\cos\theta = \hat n_x \cdot \hat n_y\)$. The **angular loss** is given by
>
> $$
> L_{\text{angle}} = 1 - \cos\theta = 2 \sin^{2}\Bigl(\tfrac{\theta}{2}\Bigr),
> $$
>
> which equals 0 when directions coincide and 2 when they are opposite.
>
> ---
>
> ### **Minor Comments**
>
> * We will explicitly define **IPC** as “**images per class**” at its first appearance.
> * The duplicate citations [23] and [24] refer to the same source; we will merge them in the final manuscript.
>
> ---
>
> We hope that these detailed clarifications convincingly address your concerns. Thank you again for your positive assessment and constructive suggestions.
>
> ---
>
> **Reference**
>
> [1] *Diversified Semantic Distribution Matching for Dataset Distillation*.

---

### Note · Authors · 2025-08-14

We sincerely thank the reviewers and AC for their thoughtful feedback and constructive discussion. This paper introduces Hyperbolic Dataset Distillation (HDD), which reformulates distribution matching in Lorentzian hyperbolic space to inject a hierarchical inductive bias. By aligning hyperbolic centroids, HDD emphasizes prototypical samples while suppressing noisy or boundary cases, and enables hierarchical pruning of real data. The method is simple, plug-and-play, and consistently improves multiple strong baselines across diverse datasets.

**Clarifications & Results.**

* **Cross-entropy compatibility:** In hybrid methods (e.g., IDM, DSDM), the CE term remains in Euclidean space to ensure consistency with standard classifier heads.
* **Extensibility:** We integrated HDD into DSDM and IID (DM), beyond the two baselines in the paper, and observed consistent gains, underscoring generality.
* **Prototypicality:** Samples closer to the hyperbolic origin are visually cleaner and more centered; far-origin samples are often occluded or cluttered, supporting the hierarchy interpretation.
* **Robustness & efficiency:** Ablations over curvature/scaling show stable performance even at extreme values. Runtime overhead is moderate (e.g., +1.7s per 100 iterations on CIFAR-10 with batch size 256) and memory is virtually unchanged. Tangent-space and exact Fréchet means perform comparably; we adopt the former for efficiency.
* **Scalability:** HDD remains effective on higher-resolution ImageNet-subset data (224×224), and we plan extensions to text and audio.
* **Theoretical perspective:** Centroid alignment constitutes first-order (moment) matching. In non-positive curvature spaces, barycentric maps are non-expansive w\.r.t. Wasserstein distance, linking centroid discrepancy to distributional gap. While full higher-order analysis is ongoing, our empirical evidence shows HDD consistently yields closer alignment than Euclidean DM.

**Planned Revisions.** We will unify notation, define IPC at first use, merge duplicate references, add missing ablations and visualizations, and release code. Appendix space will detail the distinction between tangent-space and exact centroid computation.

With these clarifications and additional experiments, reviewers acknowledged the method’s empirical stability, simplicity, and extensibility. We believe that the main concerns have been addressed, and the quality of our paper has been further strengthened.

---

### Decision · Program_Chairs · 2025-09-17

**Decision:**

Accept (poster)

**Comment:**

This paper proposes hyperbolic dataset distillation. Distribution matching is currently restricted to Euclidean space and this paper overcomes that gap in literature. After the review and rebuttal stage, the paper received an average rating of 4, based on one accept (5), two weak accepts (4), and one weak reject (3).

Regarding the strengths of the paper, the reviewers are in agreement. All positively not the novelty of the paper, the writing, and the empirical outcomes. Reviewer 9awn (score 5) asked a few prodding questions on the generalization capabilities of the method, for which the authors provide extra empirical results. Reviewer gver (score 4) has multiple good points regarding missing information on ablations and scalability. Here too, the authors provide the required answers, backed up with empirical outcomes. Reviewer arwV (score 4) had very similar concerns. Lastly, reviewer RBVi (score 3) pointed out in the final internal discussions that they had concerns regarding a lack of theoretical analyses. They find the final response by the authors interesting and insightful. While the reviewer would still like to know more precisely whether Riemannian mean matching indeed corresponds to the distribution matching (beyond bounds). However, they point out that based on the novelty of the paper, they are not opposed to acceptance.

As such, the AC follows the reviewer consensus and votes for acceptance. The strengths of the paper are clear and the weaknesses mostly concerned technical questions and additional insights/ablations, which have largely been addressed.